# Low carbon economic dispatch of integrated energy system based on coupled operation of OCPP-P2G-CHP

Jingjing Ma ®*, Wentao Huang, Can Lv, Fan Liu, Jun He, Yukun Yang

Hubei Key Laboratory for High-Efficiency Utilization of Solar Energy and Operation Control of Energy Storage System, Hubei University of Technology, Wuhan, China

* 1367595278@qq.com

## Abstract

In the context of the "double carbon" initiative, the exploration of strategies for energy conservation and emission reduction through both policy and technological approaches has emerged as a significant area of contemporary research. Based on this, a low-carbon economic dispatch model of the integrated energy system (IES) including oxygen-enriched combustion power plant (OCPP), two-stage power-to-gas (P2G), and combined heat and power (CHP) units is constructed. The objective of the model is to optimize total operational expenditures while simultaneously improving carbon resource recovery and mitigating emissions. Notable advancements encompass the transformation of traditional thermal power units into OCPP, the formulation of mathematical models for OCPP, the implementation of two-stage P2G systems, and CHP systems, as well as the incorporation of a demand response (DR) mechanism. The findings from the simulation demonstrate that the OCPP-P2G-CHP integrated system exhibits a marked enhancement in carbon capture efficiency when juxtaposed with conventional systems. Additionally, it results in a 7.8% reduction in overall costs and a 30.2% decrease in carbon emissions. This research substantiates the viability and innovative nature of the proposed model, underscoring its potential for scalable application within industrial and urban energy networks.

## Introduction

In light of the pressing demands for economic development and the escalating environmental challenges faced by numerous nations, it has become untenable to meet the material needs of the population through reliance on a singular energy source within the energy framework. Consequently, there is a growing emphasis on the development of IES [1,2]. Through the incorporation of diverse resource types, integrated energy systems are capable of addressing a range of requirements and

**Data availability statement:** All relevant data are within the manuscript and its Supporting Information files.

**Funding:** The author(s) received no specific funding for this work.

**Competing interests:** The authors have declared that no competing interests exist.

**Abbreviations:** IES, Integrated energy system; OCPP, Oxygen-enriched combustion power plant; P2G, Power-to-gas; CHP, Combined heat and power; DR, Demand response; ASU, Air separation unit; OST, Oxygen storage tank; CC, Carbon capture; HFC, Hydrogen fuel cells; GB, Gas boilers; EB, Electric boilers; GS, Gas storage; EL, Electrolysis; MR, Methanation reaction; WP, Wind power; PV, Photovoltaic

facilitating holistic development [3,4]. The notion of IES encompasses more than merely supplying a singular energy source. It is defined by the cohesive integration and synergistic interaction of various energy sources, including cooling, heating, and electricity. This framework has attracted considerable attention in relation to the optimization of energy dispatch [5,6], reliability assessment [7,8], and the development of evaluation metrics [9], among other areas. The optimal allocation of IES represents a critical component of system design. Nevertheless, within the framework of the ongoing global transition towards low-carbon energy, it is increasingly inadequate to prioritize economic considerations alone in the integrated energy system. It is essential to incorporate low-carbon objectives as a fundamental criterion in the development of IES [10].

In order to achieve decarbonization of the energy system and satisfy the electrical energy requirements of consumers, CHP systems have been extensively implemented [11]. Wei et al. [12] examines the issue of multi-objective cooperative control of CHP units, focusing on the enhancement of flexibility to meet various control requirements. Zhijun et al. [13] addressed a multi-objective optimization challenge associated with a CHP optical storage thermal power system through the application of a mixed-variable objective decomposition model. This approach facilitates the economically coordinated scheduling of both power and thermal systems, thereby enhancing energy efficiency. Dawei et al. [14] examines the decision-making challenges associated with CHP units engaged in the intraday flexibility trading market. He introduces a self-scheduling strategy aimed at augmenting the real-time flexibility of these CHP units, resulting in an increase in overall profitability and demonstrating significant practical applicability. Collectively, these studies primarily emphasize the economic dimensions of the system, with insufficient attention given to the low-carbon attributes.

In IES, P2G technology enhances the efficiency of renewable energy utilization by transforming surplus electricity into natural gas for storage purposes [15]. P2G serves as a crucial electric coupling mechanism; it not only facilitates an increased consumption rate of renewable energy by utilizing excess energy [16], but it also possesses substantial potential for reducing carbon emissions, thereby enabling participation in carbon trading markets [17]. Z et al. [18] emphasizes the synchronized functioning of green multi-energy ship microgrids and introduces a novel two-phase coordinated operational framework along with associated methodologies aimed at achieving cost-effective and low-carbon maritime transportation. Yang et al. [19] introduced a multi-stage coordinated scheduling approach designed to address the uncertainties inherent in electric-hydrogen integrated energy systems. His findings offer significant contributions to the future advancement of this system, particularly regarding optimal scheduling and system planning. Hydrogen, as an intermediate product, serves as a clean energy source with extensive applications in industrial conversion [20], fuel cells [21], IES [22, 23], and transportation [24]. Hydrogen presents significant advantages as a low-carbon and environmentally sustainable energy source, with the capability for large-scale storage over extended durations. It is anticipated that hydrogen will emerge as a prominent clean energy alternative in the future [25].

While the aforementioned studies contribute valuable insights regarding renewable energy utilization and the adaptability of energy systems, they fail to address the role of oxygen in the dispatching process.

The advancement and implementation of carbon capture technology represent a significant strategy for achieving reductions in carbon emissions. Traditional carbon capture systems designed for the capture of flue gases from combustion processes exhibit limitations, including low capture efficiency and elevated energy requirements. However, recent developments in oxygen-enriched combustion capture (OCC) technology have the potential to address these deficiencies. OCC utilizes oxygen-enriched combustion in place of air, thereby producing flue gases with a higher concentration of CO2 and enhancing the overall efficiency of carbon capture processes [26]. OCC demonstrates a strong alignment with conventional coal-fired power generation technologies, which has resulted in a significant level of acceptance within the industry [27]. Wenwe et al. [28] initiated a pioneering research endeavor focused on improving the efficiency of wind energy utilization while concurrently reducing carbon emissions. He proposed a novel methodology for optimizing the allocation of low-carbon energy resources in OCPP, leveraging wind energy to produce oxygen. Yunyun et al. [29] has developed a comprehensive electric-heat-gas energy system that incorporates coupled OCC-P2G operations and hydrogen-doped gas technology. This system is designed to enhance energy integration while prioritizing the reduction of carbon emissions and the optimization of energy utilization. Meanwhile, Hu et al. [30] undertook a comparative analysis of carbon capture process simulations and performance evaluations of two pressurized oxygen-enriched combustion power plants, focusing on the effects of oxygen-enriched combustion on their operational efficiency. While these studies provide significant insights into the synergistic interplay between oxygen-enriched combustion and P2G technology, they do not adequately address the critical role of CHP systems within an integrated energy framework.

In conclusion, the literature indicates that oxygen-enriched combustion plants exhibit significant energy consumption and incur high costs associated with oxygen production. To mitigate these challenges and facilitate the low-carbon economic operation of integrated energy systems while also utilizing carbon dioxide, this study introduces a low-carbon economic dispatch strategy that incorporates the synergistic operation of OCPP, P2G technology, and CHP systems. Initially, conventional thermal power units will be upgraded with oxygen-enriched combustion technology. Subsequently, the integrated operation of the oxygen-enriched combustion power plant will be analyzed in conjunction with P2G technology and CHP systems. The CHP will supply a portion of the energy required for the air separation unit (ASU) and carbon capture (CC) processes of the OCPP, while the P2G system will contribute some of the oxygen needed for the OCPP, thereby reducing the energy demands of the ASU and enhancing the operational efficiency of the OCPP. Finally, the proposed coupled OCPP-P2G-CHP operational model is validated through simulation, demonstrating its effectiveness and feasibility. A flowchart of the proposed approach is illustrated in Fig 1.

The main contributions of this paper are summarized below:

(1) This study involves the conversion of conventional thermal power units into OCPP and the development of models for OCPP, two-stage P2G, and CHP systems, along with their interconnection mechanisms. An IES model is established, utilizing the OCPP-P2G-CHP coupling operational mode.

(2) A demand response (DR) mechanism is incorporated into the modeling framework, leading to the formulation of a low-carbon economic dispatch model for the IES that accounts for the coupled operation of OCPP, P2G, and CHP systems in conjunction with the DR mechanism.

(3) The DR costs are integrated into the overall system cost, with the objective of minimizing the total system cost. Five distinct scenarios are devised to analyze the output and carbon emissions of each component within the IES.

Ultimately, the dispatch results from each scenario are compared and analyzed to assess the feasibility and advantages of the proposed model presented in this paper. The rest of the paper is organized as follows. Section 1 presents the IES model considering the coupled OCPP-P2G-CHP operation. Section 2 establishes the low carbon economy dispatch

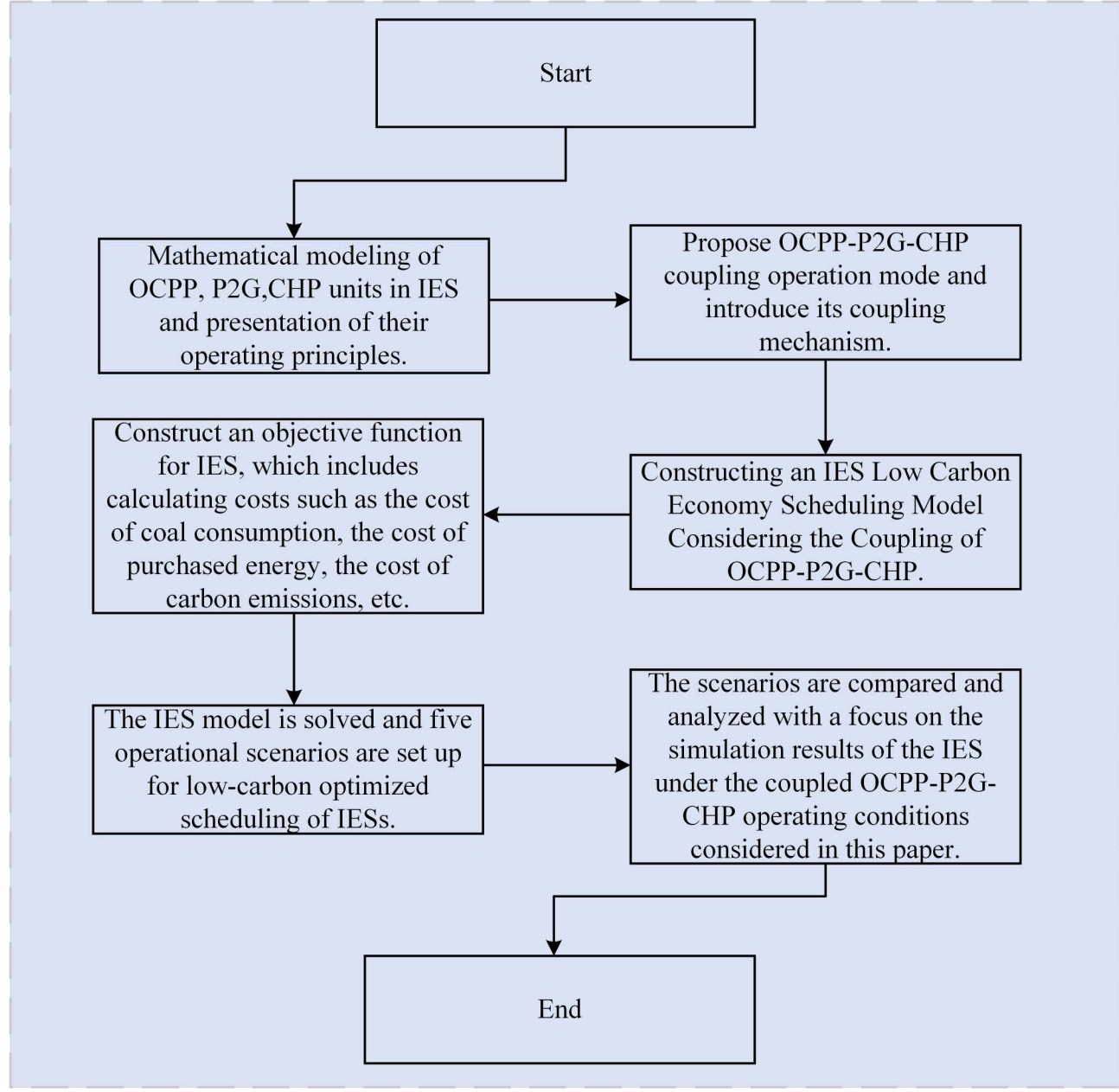

**Fig 1. Flowchart of the proposed approach.**

model for IES. Section 3 solves the dispatch model. Section 4 analyzes the case studies under different operation modes. Section 5 summarizes the paper.

## 1. IES dispatching structure considering OCPP-P2G-CHP coupling

Fig 2 illustrates the overarching structure of the IES. The electrical load demand is mainly served by OCPP, photovoltaics, wind power, CHP, and hydrogen fuel cells (HFC). The thermal load is served by gas boilers (GB), HFC, electric boilers (EB), and thermal network. The gas load is served by gas storage (GS), P2G units, and gas network.

## 1.1. OCPP modeling

**1.1.1. Oxygen-enriched combustion technology.** Oxygen-enriched combustion involves the utilization of air with an oxygen concentration exceeding 21% as the combustion medium. This process results in the generation of flue gas with a significantly elevated concentration of carbon dioxide, facilitating its efficient capture [31]. Empirical studies have demonstrated that the implementation of oxygen-enriched combustion in coal-fired systems enhances combustion efficiency, optimizes energy utilization, lowers operational costs, and increases the stability of unit operations. Consequently, the conversion of conventional thermal power plants into oxygen-enriched combustion facilities holds significant research potential.

**1.1.2. OCPP structure.** Fig 3 illustrates the configuration of an oxygen-enriched combustion power plant, which incorporates a carbon capture system designed for oxygen-enriched combustion. This system comprises a CC unit, an ASU, and an OST. The design targets the retrofitting of a conventional small-scale thermal power unit for oxygen-enriched combustion. The CC unit is designed to extract $CO_2$ from the flue gases generated during an oxygen-enriched combustion process. This unit requires an input of power to function and subsequently outputs the captured $CO_2$. The $CO_2$ that is

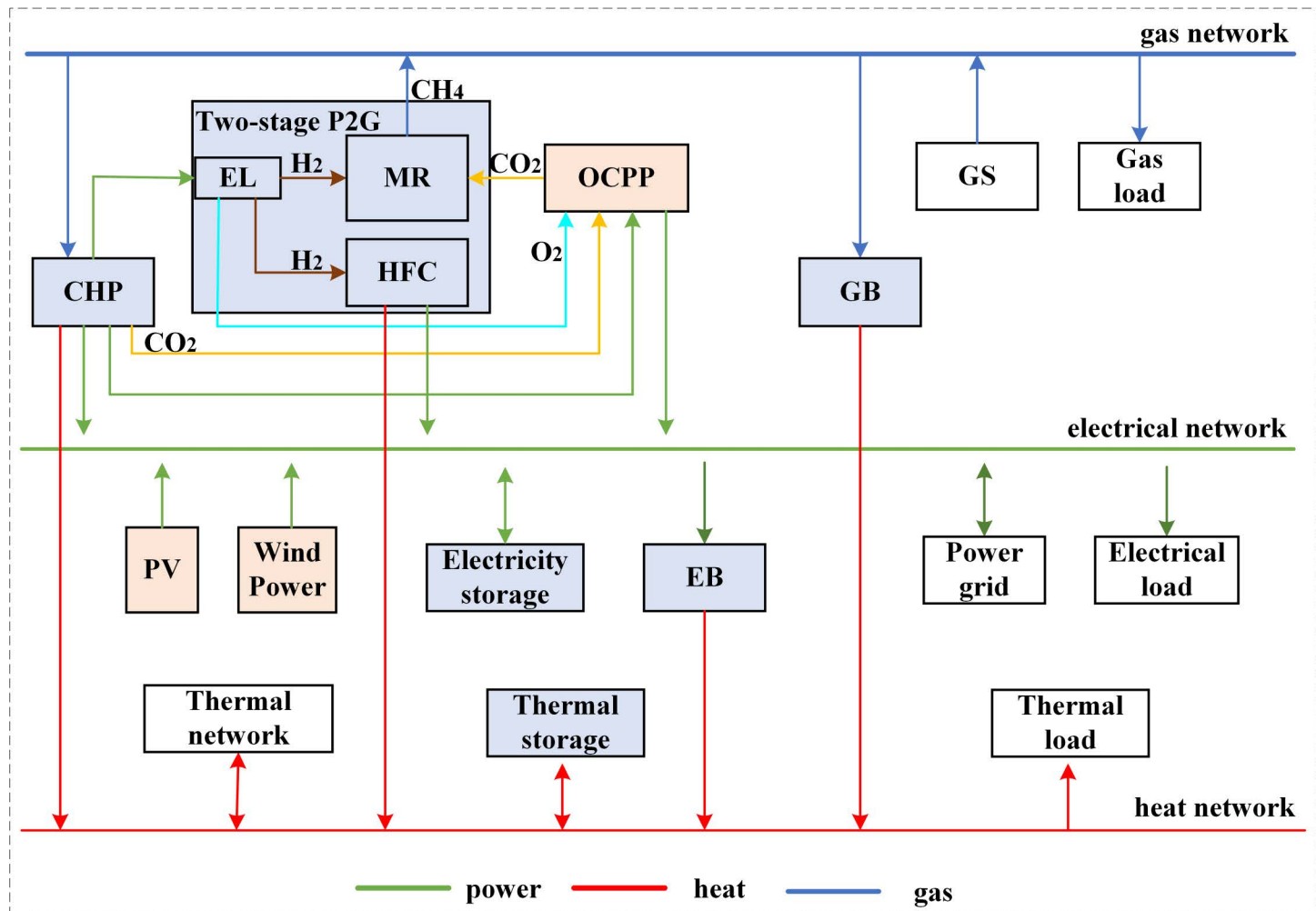

**Fig 2. IES frame diagram.**

---

captured serves as a feedstock for the second-stage reaction within the P2G plant, thereby facilitating the comprehensive utilization of $CO_2$. In contrast to post-combustion carbon capture facilities, oxygen-enriched plants are equipped with an OST, facilitating efficient energy utilization over time and enabling flexible output adjustments.

During operation of the OCPP, a small portion of the CO2 goes to the atmosphere and most of it is used in the methane reaction process of the P2G plant. The mathematical modeling of the OCPP and CC is as follows [32]:

$$
\begin{cases}
P_t^G = P_t^{GN} + P_t^{GC} \\
P_t^{GC} = P_t^{AC} + P_t^{CC} \\
C_t^{ALL} = \alpha_G \left( P_t^G + P_t^{CHP} \right) \\
C_t^N = C_t^{ALL} - C_t^{CC} \\
V_t^G = \chi_G P_t^G
\end{cases}
\tag{1}
$$

Where $P_t^G$ is the total output for OCPP, $P_t^{GN}$ is the net output of OCPP, $P_t^{GC}$ is the operational energy consumption of OCPP, $P_t^{CC}$ is the power consumption provided to CC by OCPP, $P_t^{AC}$ is the energy consumption of OCPP supplied to the ASU, $C_t^{ALL}$ is the $CO_2$ emission of the system unit; $\alpha_G$ is the generation factor; $P_t^{CHP}$ is the power generated by CHP, $C_t^N$ is the net $CO_2$ emissions from OCPP, is the amount of oxygen required for OCPP, $\chi_G$ is the oxygen consumption factor.

Following the treatment of flue gas from the thermal power unit through desulfurization, denitrification, and dehydration processes, the concentration of CO2 in the flue gas exceeds 80%. This elevated CO2 content results in a reduced energy

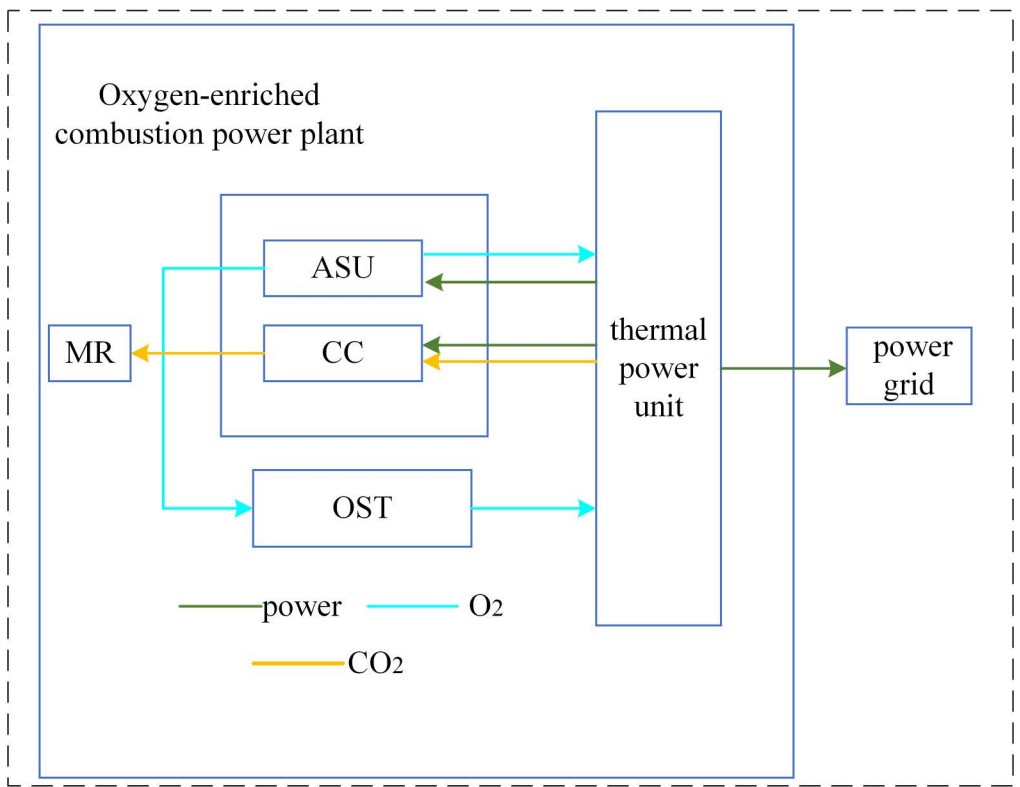

**Fig 3. Structure of OCPP.**

consumption for CO2 capture per unit of carbon capture (CC), with capture efficiencies reaching as high as 98%. The modeling of this process is presented as follows [33]:

$$\begin{cases} P_t^{CPU} = \alpha_{cc} C_t^{CC} \\ C_t^{CC} = \varphi_t^{CC} e_{TPU} P_t^{CC} \\ C_{min}^{CC} \leq C_t^{CC} \leq C_{max}^{CC} \\ C_t^{OCPP} = (1 - \varphi_t^{CC}) e_{TPU} P_t^{CC} \\ 0 \leq \varphi_t^{CC} \leq \varphi_{max}^{CC} \end{cases} \tag{2}$$

Where $P_t^{CPU}$ is the total operating energy consumption of the CC, $C_t^{CC}$ is the amount of carbon captured by the CC, $\alpha_{cc}$ is the energy factor for capture, $\varphi_t^{CC}$ is the carbon capture intensity factor for CC at time period t, $e_{TPU}$ is the carbon intensity factor for thermal power units, $C_t^{OCPP}$ is the amount of CO2 emitted into the atmosphere by the thermal power unit at time period t.

The ASU utilizes electrical energy to extract oxygen from the atmosphere. Given the extended duration required for both initiation and cessation of the ASU's operations, coupled with its significant energy demands—where the minimum energy consumption is 50% of the rated power—this study focuses exclusively on its operational phase. The corresponding model is as follows:

$$\begin{cases} V_t^{ASU} = \alpha_{ASU} P_t^{ASU} \\ a_{ASU,min} P_{max}^{ASU} \leq P_t^{ASU} \leq P_{max}^{ASU} \end{cases} \tag{3}$$

Where $V_t^{ASU}$ is the oxygen production of ASU, $\alpha_{ASU}$ is the oxygen production per unit power, $P_t^{ASU}$ is the total ASU operational energy consumption, $a_{ASU,min}$ is the minimum operating factor of the ASU, $P_{max}^{ASU}$ is the upper limit of energy consumption for the operation of the ASU.

To mitigate the phenomenon of "oxygen-dependent power generation" within the context of the OCPP, which refers to the efficient utilization of oxygen while enhancing the electrical output of oxygen-rich thermal power units, the integration of an OST is proposed. During low load conditions, the ASU is capable of continuously producing oxygen to facilitate oxygen-enriched combustion. Additionally, the P2G system generates oxygen through the electrolysis of water, resulting in surplus oxygen production. To optimize the utilization of oxygen resources, the excess oxygen generated by both the ASU and P2G can be stored in the OST. In periods of high load, the oxygen-enriched combustion unit prioritizes fulfilling electrical load demands, which consequently reduces the oxygen output from the ASU. During these times, the oxygen previously stored in the OST during low load periods can be introduced into the combustion chamber, thereby ensuring the proper functioning of the oxygen-enriched combustion unit and maintaining a consistent electricity supply.OST stores oxygen, and oxygen generation and transfer are modeled as follows:

$$V_t^{OST} = V_{t-1}^{OST} + V_t^{OSC} - V_t^{OSD} \tag{4}$$

Where $V_t^{OST}$ is the OST oxygen volume stored, $V_t^{OSC}$ is the input oxygen volume for OST, $V_t^{OSD}$ is the oxygen volume output from OST.

## Two-stage modeling of P2G conversion

P2G technology facilitates the conversion of electricity into natural gas, thereby allowing for bidirectional energy transfer between grids. This process leverages existing natural gas infrastructure to support large-scale distribution and utilization, thereby promoting the off-site consumption of renewable energy sources.

In order to provide a precise representation of the P2G system, this study substitutes the conventional P2G model with electrolysis (EL), methanation reaction (MR), and HFC, subsequently refining it into two distinct processes: the production of hydrogen through electrolysis and the methanation process. The electrolysis of water is conducted in the EL phase, where a portion of the generated hydrogen energy is utilized in the MR for the methanation reaction, while the remaining hydrogen is directly converted into electricity and heat via HFC, thereby facilitating the efficient utilization of hydrogen energy. The mathematical model for the first stage of electrohydrogenation in the P2G system is presented as follows [34]:

$$\begin{cases} V_t^{H_2} = \eta_{H2} P_t^{P2G} \\ V_t^{O_2} = \eta_{H2,O2} V_t^{H_2} \\ \lambda_1^{EL} \le P_{t+1}^{P2G} - P_t^{P2G} \le \lambda_u^{EL} \end{cases} \tag{5}$$

Where $V_t^{H_2}$ is the hydrogen production, $\eta_{H2}$ is the Hydrogen conversion efficiency, $P_t^{P2G}$ is the P2G Input Power, $V_t^{O_2}$ is the Oxygen production; $\eta_{H2,O2}$ is the coefficient of reaction between $H_2$ and $O_2$ in the electrolysis of water, $\lambda_u^{EL}$ and $\lambda_1^{EL}$ are the upper and lower limits of EL respectively.

HFC recovers excess $H_2$ from water electrolysis in the first stage of the P2G process. The electrical and thermal power output of the HFC is as follows:

$$\begin{cases} P_t^{HFC} = \eta_{HFC}^e H_t^{HFC} \\ R_t^{HFC} = \eta_{HFC}^r H_t^{HFC} \\ H_{min}^{HFC} \le H_t^{HFC} \le H_{max}^{HFC} \\ \lambda_1^{HFC} \le H_{t+1}^{HFC} - H_t^{HFC} \le \lambda_u^{HFC} \\ K_{min} \le \dfrac{R_t^{HFC}}{P_t^{HFC}} \le K_{max} \end{cases} \tag{6}$$

Where $H_t^{HFC}$ is the hydrogen power input to the HFC at time t, $P_t^{HFC}$ and $R_t^{HFC}$ are the electric and thermal power generated by the HFC at time t, $\eta_{HFC}^e$ and $\eta_{HFC}^r$ are the electric and thermal conversion efficiencies of the HFC, $H_{min}^{HFC}$ and $H_{max}^{HFC}$ are the upper and lower limits of the hydrogen power input to the HFC, $\lambda_u^{HFC}$ and $\lambda_1^{HFC}$ are the climbing upper and lower limits of HFC, $K_{max}$ and $K_{min}$ are the upper and lower limits of the thermoelectric ratio of HFC.

The second stage takes place in a methanation unit, where $H_2$ and $CO_2$ undergo a methanation reaction to synthesize methane ($CH_4$) and water. The mathematical model of the second stage is given below:

$$\begin{cases} G_t^{MR} = \eta_{MR} H_t^{MR} \\ C_t^{MR} = \sum\limits_{t=1}^{T} \alpha_G G_t^{MR} \\ \lambda_1^{MR} \le H_{t+1}^{MR} - H_t^{MR} \le \lambda_u^{MR} \end{cases} \tag{7}$$

Where $G_t^{MR}$ is the gas production power of the methanization reaction device at time t, $\eta_{MR}$ is the methanization efficiency, $H_t^{MR}$ is the hydrogen consumption power of the methanization reaction unit at time t, $C_t^{MR}$ is the amount of $CO_2$ required for the methanation reaction, $\lambda_u^{MR}$ and $\lambda_1^{MR}$ are the upper and lower limits of MR climbing, respectively.

In this paper, we consider that the amount of $CO_2$ required for the second stage of methanation under OCPP-P2G-CHP coupling conditions is equal to the amount of carbon captured in OCPP. The mathematical model is:

$$C_t^{MR} = C_t^{CC} \tag{8}$$

## Modeling of OCPP-P2G-CHP coupled system

In conclusion, the aforementioned study integrates the OCPP, P2G, and CHP systems into a comprehensive coupling model known as the OCPP-P2G-CHP model. The initial phase of the P2G technology, which focuses on the production of electric hydrogen, generates oxygen as a byproduct. This oxygen is subsequently introduced into the OCPP, facilitating the complete combustion of coal powder in an oxygen-enriched environment, thereby yielding flue gas with a high concentration of carbon dioxide that is more amenable to capture. The synergistic operation with the OCPP further mitigates the carbon emissions of the overall system. Additionally, the carbon dioxide captured by the OCPP can be utilized in the second phase of the P2G process, specifically during the methanation reaction, which effectively lowers both the costs associated with carbon dioxide sequestration and the expenses related to the procurement of natural gas. Furthermore, the integration with CHP diminishes the direct carbon dioxide emissions released into the atmosphere from the CHP system, while simultaneously allowing CHP to provide power to the OCPP. This arrangement leads to a reduction in the output of thermal units within the OCPP, thereby contributing to a decrease in carbon emissions.

The operational characteristics of the coupled OCPP-P2G-CHP system are illustrated in Fig 4. The coupling of OCPP-P2G-CHP facilitates the utilization of $CO_2$, $H_2$ and $O_2$ and reduces pollution. By considering the combined operation, it becomes feasible to decrease the output of the ASU in the OCPP while increasing the output of the CC system, thereby facilitating greater carbon capture and promoting a cleaner and more efficient IES.

The CHP, P2G, and OCPP coupling is modeled as follows:

$$\begin{cases} P_t^{CHP} = P_t^{P2G} + P_t^{CHC} + P_t^{CHA} + P_t^{E} \\ P_t^{CPU} = P_t^{CC} + P_t^{CHC} \\ P_t^{ASU} = P_t^{AC} + P_t^{CHA} \end{cases} \tag{9}$$

Where $P_t^{E}$ is the power supplied by the CHP to the grid, $P_t^{CHC}$ is the operating energy consumption of the CHP for CC, and $P_t^{CHA}$ is the operating energy consumption of the CHP for the ASU.

The OCPP is modeled as follows after the OCPP is operated in combination with P2G:

$$V_t^{O2} + V_t^{ASU} = V_t^{G} \tag{10}$$

## 2 Low-carbon economic dispatch model for IES considering OCPP-P2G-CHP coupling

### 2.1. Equipment Modeling

**2.1.1. CHP Modeling.** CHP systems facilitate the concurrent provision of electrical and thermal energy through the combustion of natural gas. Nevertheless, the conventional operational approach of "determining power output based on thermal demand" imposes limitations on the electrical output of the CHP unit, thereby hindering its overall efficiency. In contrast, CHP systems that incorporate an adjustable heat-to-power ratio can optimize the energy supply balance between thermal and electrical loads, thereby enhancing the functionality of the unit. Waste heat boilers utilized in CHP systems are classified into two categories: those without supplementary combustion and those with supplementary combustion. The thermoelectric efficiency of a boiler lacking supplementary combustion is intrinsically linked to the performance of the unit and is typically maintained at a constant level. In contrast, a boiler equipped with supplementary combustion allows for adjustments to be made, thereby enhancing the overall energy utilization efficiency through the modulation of supplementary combustion levels. Consequently, a CHP unit featuring a supplementary-fired boiler possesses the capability to modify its power and heat output in response to real-time demands for power and heat, thereby further optimizing operational efficiency. The mathematical representation of this system is as follows:

$$\begin{cases} P_t^{CHP} = \delta_{CHP,e} G_t^{CHP} \\ Q_t^{CHP} = \delta_{CHP,h} G_t^{CHP} \\ G_{min}^{CHP} \leqslant G_t^{CHP} \leqslant G_{max}^{CHP} \\ \Delta G_{min}^{g} \leqslant G_{t-1}^{g} - G_t^{g} \leqslant \Delta G_{max}^{g} \\ \lambda_{min} \leqslant Q_t^{CHP}/P_t^{CHP} \leqslant \lambda_{max} \end{cases} \tag{11}$$

Where $P_t^{CHP}$ is the output electric power of CHP,$Q_t^{CHP}$ is the output thermal power of CHP, $\delta_{CHP,e}$ and $\delta_{CHP,h}$ are the electrical and thermal efficiencies of conversion,$G_t^{CHP}$ is the input natural gas power of CHP, $G_{max}^{CHP}$ and $G_{min}^{CHP}$ are the upper and lower limits of CHP input natural gas power, $\Delta G_{max}^{g}$ and $\Delta G_{min}^{g}$ represent the upper and lower boundaries of the climbing constraints, $\lambda_{max}$ and $\lambda_{min}$ represent the maximum and minimum thresholds of the adjustable thermoelectric ratio, Thermoelectric ratio taking reference [35].

**2.1.2. GB modeling.** The GB functions by converting chemical energy into thermal energy through the combustion of natural gas, making it a widely utilized heating apparatus within IES. The thermal power output of the gas boiler is detailed as follows:

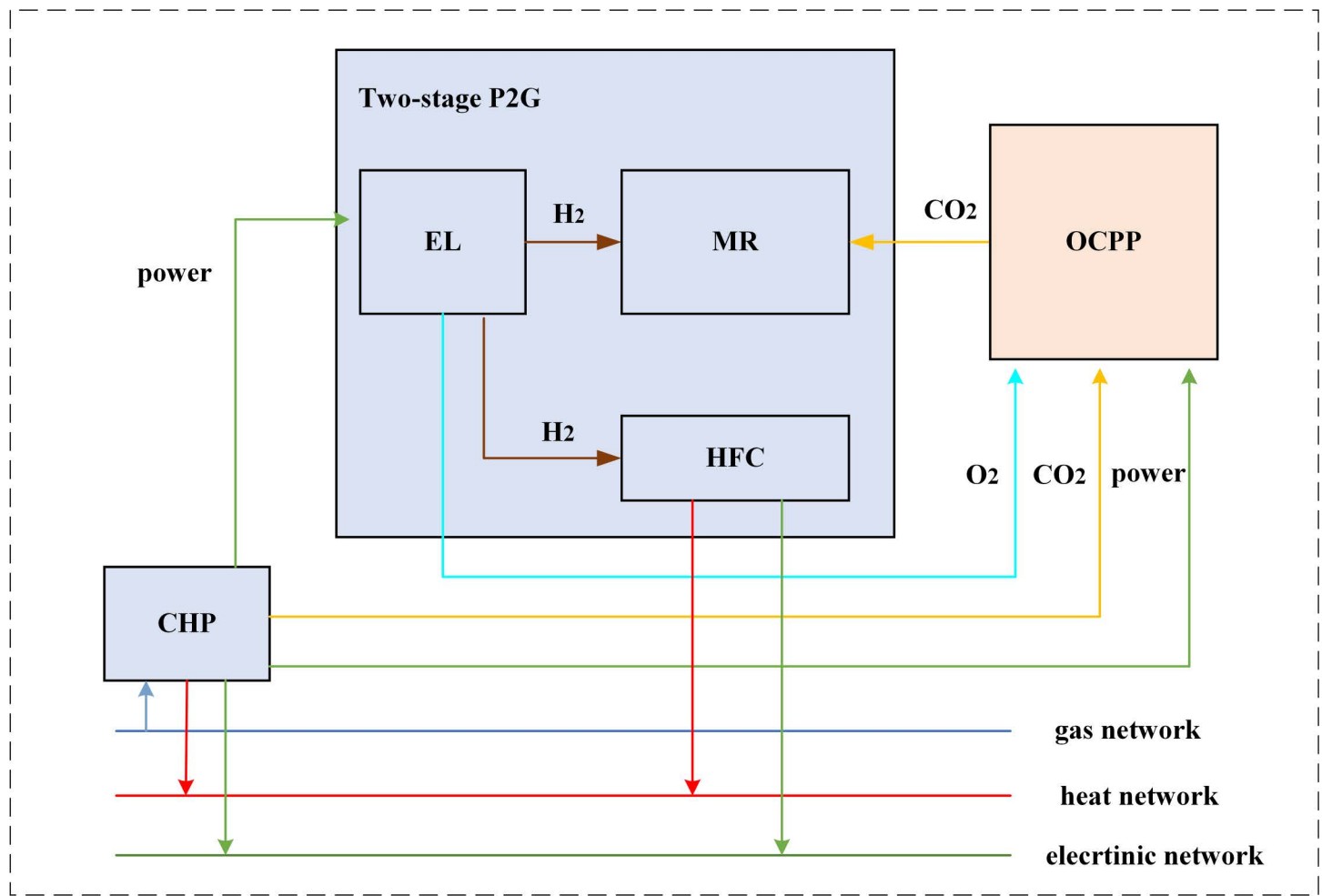

**Fig 4. The OCPP-P2G-CHP coupling relationship.**

$$\begin{cases} Q_t^{GB} = \delta_{GB} G_t^{GB} \\ Q_{min}^{GB} \leq Q_t^{GB} \leq Q_{max}^{GB} \end{cases} \quad (12)$$

Where $Q_t^{GB}$ is the output thermal power of the GB, $\delta_{GB}$ is the heating efficiency of the GB, $G_t^{GB}$ is the input natural gas power of the GB, $Q_{max}^{GB}$ and $Q_{min}^{GB}$ are the upper and lower limits of the output thermal power respectively.

**2.1.3. EB modeling.** Electric Boilers (EB) employ electrical energy to generate heat through the circulation of steam and high-temperature water vapor at specified temperatures. EB systems are characterized by their lack of emissions of pollutants, including carbon dioxide and sulfur compounds, and they exhibit a high conversion efficiency, which can exceed 95%. The thermal power output of EB is as follows [36]:

$$\begin{cases} Q_t^{EB} = \delta_{EB} P_t^{EB} \\ Q_{min}^{EB} \leq Q_t^{EB} \leq Q_{max}^{EB} \end{cases} \quad (13)$$

Where $Q_t^{EB}$ is the output thermal power of the EB at time t, $\delta_{EB}$ is the heating efficiency of EB, $P_t^{EB}$ is the input power of the EB at time t; $Q_{max}^{EB}$ and $Q_{min}^{EB}$ are the upper and lower limits of the thermal power output of the EB, respectively.

**2.1.4. Multi-component energy storage modeling.**

$$\begin{cases} E_t^n = E_{t-1}^n + \eta_{n,in} P_t^{n,in} - P_t^{n,out}/\eta_{n,out} \\ E_{min}^n \leqslant E_t^n \leqslant E_{max}^n \\ 0 \leqslant P_t^{n,in} \leqslant B_t^{n,in} P_{max}^{n,in} \\ 0 \leqslant P_t^{n,out} \leqslant B_t^{n,out} P_{max}^{n,out} \\ 0 \leqslant B_t^{n,in} + B_t^{n,out} \leqslant 1 \end{cases} \quad (14)$$

Where $n$ is the energy type, $n \in \{e, h\}$, $E_t^n$ is the amount of energy stored in the n-type energy storage device at time t, $P_t^{n,in}$ and $P_t^{n,out}$ are the stored and discharged power of the n-type energy storage device at time t, respectively. $\eta_{n,in}$ and $\eta_{n,out}$ are the storage and discharge efficiencies of the n-type energy storage device, respectively. $E_{max}^n$ and $E_{min}^n$ are the upper and lower limits of the capacity of the n-type energy storage device, respectively, $B_t^{n,in}$, $B_t^{n,out}$ are 0–1 logic variables, are the storage and discharge state of the n-type energy storage device at time t, $P_{max}^{n,in}$ and $P_{max}^{n,out}$ are the maximum power of storage and discharge of the n-type energy storage device, respectively.

## 2.2. Objective function

In order to represent the low-carbon economy of the system with respect to the OCPP-P2G-CHP coupling, this chapter develops a low-carbon economic dispatch model for the IES. The model is structured to reduce the operational expenses of the system, while simultaneously considering the functional limitations of each individual component. The objective function for optimization is defined as follows:

$$F = \min \left( f^M + f^B + f^T + f^F + f^Y + f^Q + f^D \right) \quad (15)$$

Where $f^M$, $f^B$, $f^T$, $f^F$, $f^Y$, $f^Q$, $f^D$ are the system's coal consumption cost, purchased energy cost, system's carbon emission cost, system's carbon sequestration cost, system's operating cost, system's wind, and solar energy abandonment cost, and demand response cost.

(1) Coal consumption cost

$$f^M = \sum_{t=1}^{T} [a_n (P_t^G)^2 + b_n P_t^G + c_n] \quad (16)$$

Where $a_n$, $b_n$ and $c_n$ are the constant coefficients of coal consumption characteristics of thermal power units.

(2) Purchased energy cost

$$f^B = \sum_{t=1}^{T} (\sigma_t^e P_t^{Buy} + \sigma_t^g G_t^{Buy}) \tag{17}$$

Where $\sigma_t^e$ and $\sigma_t^g$ are the price of purchased electricity and gas respectively, $P_t^{Buy}$ and $G_t^{Buy}$ are the amount of purchased electricity and gas at time t respectively.

(3) Carbon emission cost

$$\begin{cases} C = C^{Fire} + C^{Gas} \\ C^{Fire} = \delta_e \sum_{t=1}^{T} (P_t^G + P_t^{Buy}) \\ C^{Gas} = \delta_g \sum_{t=1}^{T} (Q_t^{CHP} + Q_t^{GB}) \end{cases} \tag{18}$$

Where $C$ is the amount of carbon allowances for the system, $C^{Fire}$ is the amount of carbon allowances for external grid purchases and OCPP, $C^{Gas}$ is the amount of carbon allowances for gas-fired units, $\delta_e$ and $\delta_g$ are the amount of carbon allowances for gas-fired units.

The carbon emissions of the system are as follows [37]:

$$\begin{cases} C_{Act} = C_{Act}^{Fire} + C_{Act}^{Gas} - C_{Act}^{P2G} \\ C_{Act}^{Fire} = \sum_{t=1}^{T} C_t^N + \beta_e \sum_{t=1}^{T} P_t^{Buy} \\ C_{Act}^{Gas} = \beta_g \sum_{t=1}^{T} (Q_t^{CHP} + Q_t^{GB}) \\ C_{Act}^{P2G} = \sum_{t=1}^{T} C_t^{P2G} \end{cases} \tag{19}$$

Where $C_{Act}$ is the actual carbon emission of the system,$C_{Act}^{Fire}$ is the actual carbon emission of the external grid purchased power and OCPP,$C_{Act}^{Gas}$ is the actual carbon emission of the gas-fired unit,$C_{Act}^{P2G}$ is the amount of carbon dioxide consumed by the P2G process, $\beta_e$ and $\beta_g$ are the actual carbon emission of the purchased power and the gas-fired unit, respectively.

The carbon cost of the system is as follows:

$$f^T = \sigma_c(C_{Act} - C) \tag{20}$$

Where $\sigma_c$ is the carbon emission penalty factor.

(4) Carbon sequestration cost

$$f^F = \sum_{t=1}^{T} (\sigma_f(C_t^{CC} - C_t^{P2G})) \tag{21}$$

Where $\sigma_f$ is the carbon sequestration cost factor.

(5) Operating cost

$$f^Y = \sum_{t=1}^{T} (\sigma_{wt}P_t^{wt} + \sigma_{pv}P_t^{pv} + \sigma_{gb}Q_t^{GB} + \sigma_{eb}Q_t^{EB} + \sigma_{el}P_t^{P2G}) \tag{22}$$

Where $\sigma_{wt}$, $\sigma_{pv}$, $\sigma_{gb}$, $\sigma_{eb}$, $\sigma_{el}$ are the operating cost coefficients of wind power, photovoltaic, GB, EB, and P2G, $P_t^{wt}$ and $P_t^{pv}$ are the output power of wind power and photovoltaic respectively.

(6) Wind and solar energy abandonment cost

$$f^Q = \beta_c \sum_{t=1}^{T} (P_t^{wt,q} + P_t^{pv,q})$$

(23)

Where $P_t^{wt,q}$ and $P_t^{pv,q}$ are the amount of wind and light abandoned by wind power and photovoltaic respectively, $\beta_c$ is the penalty coefficient.

(7) Demand response cost

Any transfer or curtailment of the user's load requires compensation to the user:

$$f^{DR} = \sum_{t=1}^{T} (\lambda_e^{cut} P_t^{cut} + \lambda_e^{tra} P_t^{tra} + \lambda_h^{cut} Q_t^{cut})$$

(24)

Where $\lambda_e^{cut}, \lambda_e^{tra}, \lambda_h^{cut}$ are the unit price of compensation for curtailable, transferable electric and thermal loads, respectively. $P_t^{cut}, P_t^{tra}$ are the amount of electricity that can be curtailed and transferred, $Q_t^{cut}$ is the amount of heat that can be curtailed.

## 2.3. Constraints

(1) Electrical power balance constraint [38]

$$P_t^{wt} + P_t^{pv} + P_t^{Buy} + P_t^{dis} + P_t^{GN} + P_t^{CHP} = P_t^{load} + P_t^{P2G} + P_t^{cha} + P_t^{EB}$$

(25)

Where $P_t^{wt}, P_t^{pv}$ are the power of wind power and photovoltaic grid access, $P_t^{cha}$, $P_t^{dis}$ are the charging and discharging power of electric energy storage, $P_t^{load}$ is the electric load after DR.

(2) Thermal power balance constraint

$$Q_t^{dis} + Q_t^{GB} + Q_t^{CHP} + Q_t^{EB} = Q_t^{cha} + Q_t^{load}$$

(26)

Where $Q_t^{cha}, Q_t^{dis}$ are the thermal energy storage heat storage and heat release power, $Q_t^{load}$ is the heat load after the DR.

(3) Gas power balance constraint

$$G_t^{GB} + G_t^{CHP} + G_t^{load} = G_t^{Buy} + G_t^{P2G}$$

(27)

Where $G_t^{load}$ is the gas load.

(4) Gas equilibrium constraint

$$\begin{cases} V_t^{O_2} + V_t^{OSD} + V_t^{ASU} = V_t^{OSC} + V_t^{G} \\ V_t^{H_2} = V_t^{H_2,MR} + V_t^{H_2,HFC} \end{cases}$$

(28)

Where $V_t^{H_2,MR}$ is the amount of hydrogen input to MR, $V_t^{H_2,HFC}$ is the amount of hydrogen input to HFC.

(5) OCPP constraint

$$\begin{cases} V_{min}^G \leq V_t^G \leq V_{max}^G \\ P_{min}^G \leq P_t^G \leq P_{max}^G \\ -\Delta P_G \leq P_{t+1}^G - P_t^G \leq \Delta P_G \\ 0 \leq C_t^{CC} \leq \delta_C C_t^G \\ P_{min}^{ASU} \leq P_t^{ASU} \leq P_{max}^{ASU} \end{cases} \tag{29}$$

Where $V_{max}^G, V_{min}^G$ are the max and min values of OCPP oxygen consumption, $P_{max}^G, P_{min}^G$ are the max and min values of OCPP output, $\Delta P_G$ is the climb rate of OCPP, $\delta_C$ is the max carbon capture rate, $P_{max}^{ASU}, P_{min}^{ASU}$ are the max and min values of the ASU output.

(6) P2G constraint

$$\begin{cases} P_{min}^{P2G} \leq P_t^{P2G} \leq P_{max}^{P2G} \\ H_{min}^{MR} \leq H_t^{MR} \leq H_{max}^{MR} \end{cases} \tag{30}$$

Where $P_{max}^{P2G}, P_{min}^{P2G}$ are the max and min values of the input power of P2G, $H_{max}^{MR}, H_{min}^{MR}$ are the max and min values of the hydrogen consumption power of the methanization reaction unit.

(7) Oxygen storage tank restraint

$$\begin{cases} V_{min}^{OST} \leq V_t^{OST} \leq V_{max}^{OST} \\ V_1^{OST} = V_{24}^{OST} \\ 0 \leq V_t^{OSC} \leq V_{max}^{OSC} \\ 0 \leq V_t^{OSD} \leq V_{max}^{OSD} \end{cases} \tag{31}$$

Where $V_{max}^{OST}, V_{min}^{OST}$ are the max and min values of OST oxygen content, $V_{max}^{OSC}, V_{max}^{OSD}$ are the max values of oxygen filling and oxygen discharge, respectively.

(8) Wind power and photovoltaic output constraint

$$\begin{cases} P_t^{WT} = P_t^{wt} + P_t^{wt,q} \\ P_t^{PV} = P_t^{pv} + P_t^{pv,q} \end{cases} \tag{32}$$

Where $P_t^{WT}, P_t^{PV}$ are the predicted output of wind power and PV respectively.

(9) Demand response constraint

$$\begin{cases} \left| \sum_{t=1}^T P_t^{tra} \right| \leq \tau_{tra}^e P_t^{load} \\ \sum_{t=1}^T P_t^{tra} = 0 \\ 0 \leq P_t^{cut} \leq \tau_{cut}^e P_t^{load}(t) \\ 0 \leq Q_t^{cut} \leq \tau_{cut}^h Q_t^{load}(t) \end{cases} \tag{33}$$

Where $\tau_{cut}^e, \tau_{cut}^h$ are the electric and thermal load coefficients after demand response.

## 3. Model solution

The model developed in this study is characterized as a mixed-integer nonlinear programming model, incorporating both equation and inequality constraints. The simulations were conducted using MATLAB, with the CPLEX solver employed to obtain the optimization solutions. The scheduling period is set at T = 24 hours, with a unit scheduling interval of Δt = 1 hour. The primary workflow is illustrated in Fig 5.

## 4. Example analysis

### 4.1. System parameters

Taking a comprehensive park as an example, an example analysis is carried out to verify the effectiveness of the proposed low-carbon economic dispatch model. The specifications for each piece of equipment within the system are

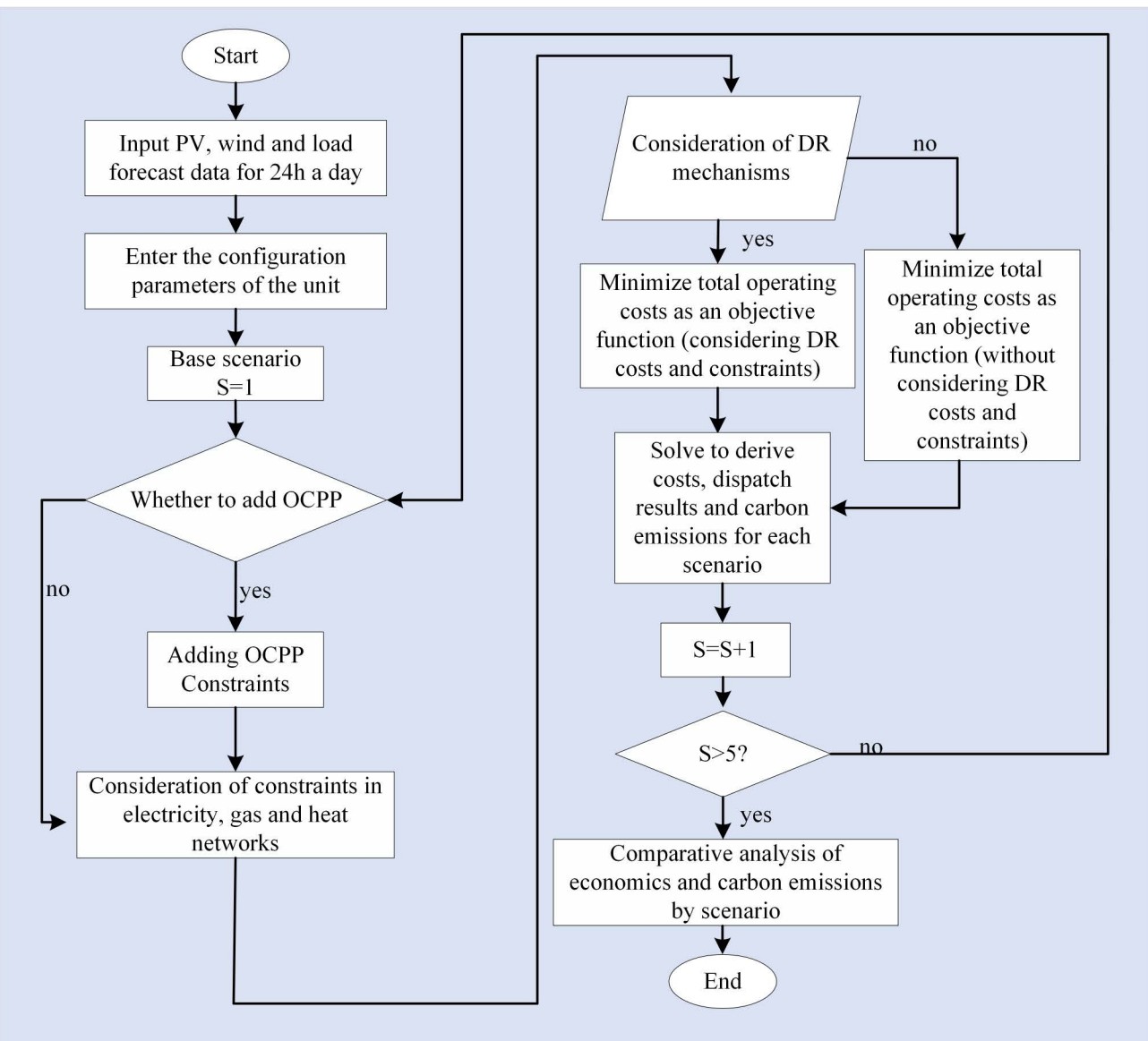

**Fig 5. Flow chart of model solving.**

presented in Tables 1 and 2. The segmented table detailing the power purchase prices is provided in Table 3. Fig 6 shows the 39-20-6 IES diagram, and the flow direction of energy is shown in Fig 2. The example is an IES consisting of a IEEE-39 node power network, a 20-node gas network, and a 6-node thermal network [39]. Additionally, the projected values for wind power generation, photovoltaic power generation, and load forecasts are illustrated in Figs 7 and 8 [40]. Forecasted figures and specific data for each energy source are shown in the supporting information in S1–S4 Figs and S1.File

**Table 1. Basic parameters of OCPP.**

| Parameter | Value | Parameter | Value |
|---|---|---|---|
| $\alpha_{cc}(kW \cdot h\ /kg)$ | 0.82 | $\Delta P_G(kW)$ | 4000 |
| $\chi_G(m^3/kW)$ | 0.4 | $\delta_C$ | 0.98 |
| $\alpha_G(t/kW \cdot h)$ | 0.98 | $\alpha_{ASU}(m^3/kW)$ | 2.4 |
| $P_{min}^G(kW)$ | 2000 | $P_{max}^G(kW)$ | 20000 |
| $V_{min}^{OST}(m^3)$ | 500 | $V_{max}^{OST}(m^3)$ | 5000 |
| $P_{min}^{ASU}(kW)$ | 500 | $P_{max}^{ASU}(kW)$ | 10000 |
| $V_{max}^{OSD}(m^3)$ | 500 | $V_{max}^{OSC}(m^3)$ | 500 |

**Table 2. Other parameters in the system.**

| Parameter | Value | Parameter | Value |
|---|---|---|---|
| $\delta_{CHP}$ | 0.4 | $\kappa_{CHP}$ | 1.2 |
| $\delta_{GB}$ | 0.6 | $\delta_{EB}$ | 0.9 |
| $\eta_{H2}$ | 0.88 | $\eta_{H2,O2}$ | 0.5 |
| $\lambda_1^{EL}$ | -4 | $\lambda_u^{EL}$ | 4 |
| $\lambda_1^{MR}$ | -5 | $\lambda_u^{MR}$ | 5 |
| $\lambda_1^{HFC}$ | -5 | $\lambda_u^{HFC}$ | 5 |
| $K_{min}$ | 0.5 | $K_{max}$ | 2.1 |
| $P_{min}^{P2G}(kW)$ | 5000 | $P_{max}^{P2G}(kW)$ | 25000 |
| $\eta_{MR}$ | 0.55 | $Q_{min}^{GB}(kW)$ | 1000 |
| $Q_{max}^{GB}(kW)$ | 10000 | $Q_{min}^{EB}(kW)$ | 500 |
| $Q_{max}^{EB}(kW)$ | 5000 | $Q_{min}^{CHP}(kW)$ | 12000 |
| $Q_{max}^{CHP}(kW)$ | 42000 | $H_{min}^{MR}(kW)$ | 2000 |
| $H_{max}^{MR}(kW)$ | 20000 | $H_{min}^{HFC}(kW)$ | 2000 |
| $H_{max}^{HFC}(kW)$ | 20000 | $a_n$ | 0.0000014 |
| $b_n$ | 0.2 | $c_n$ | 0.076 |
| $\delta_e(kg/kW)$ | 0.61 | $\delta_g(kg/kW)$ | 0.31 |
| $\beta_e(kg/kW)$ | 0.95 | $\beta_g(kg/kW)$ | 0.44 |
| $\sigma_c(¥/kg)$ | 0.2 | $\sigma_f(¥/kg)$ | 0.03 |
| $\sigma_{wt}(¥/kW)$ | 0.01 | $\sigma_{pv}(¥/kW)$ | 0.01 |
| $\sigma_{el}(¥/kW)$ | 0.03 | $\sigma_{gb}(¥/kW)$ | 0.01 |
| $\sigma_{eb}(¥/kW)$ | 0.01 | $\beta_c(¥/kW)$ | 0.2 |
| $\lambda_e^{tra}(¥/kW)$ | 0.05 | $\lambda_e^{tra}(¥/kW)$ | 0.05 |
| $\lambda_h^{cut}(¥/kW)$ | 0.1 | $\tau_{tra}^e(¥/kW)$ | 0.2 |
| $\tau_{cut}^e$ | 0.1 | $\tau_{cut}^h$ | 0.1 |

**Table 3. Purchased power and gas prices at different periods.**

| Time | Power price /(¥/(k·Wh)) | Gas price/(¥/m³) |
|---|---|---|
| 01:00—07:00 | 0.12 | 2.5 |
| 08:00—11:00 | 0.35 | 2.5 |
| 12:00—17:00 | 0.22 | 2.5 |
| 18:00—24:00 | 0.35 | 2.5 |

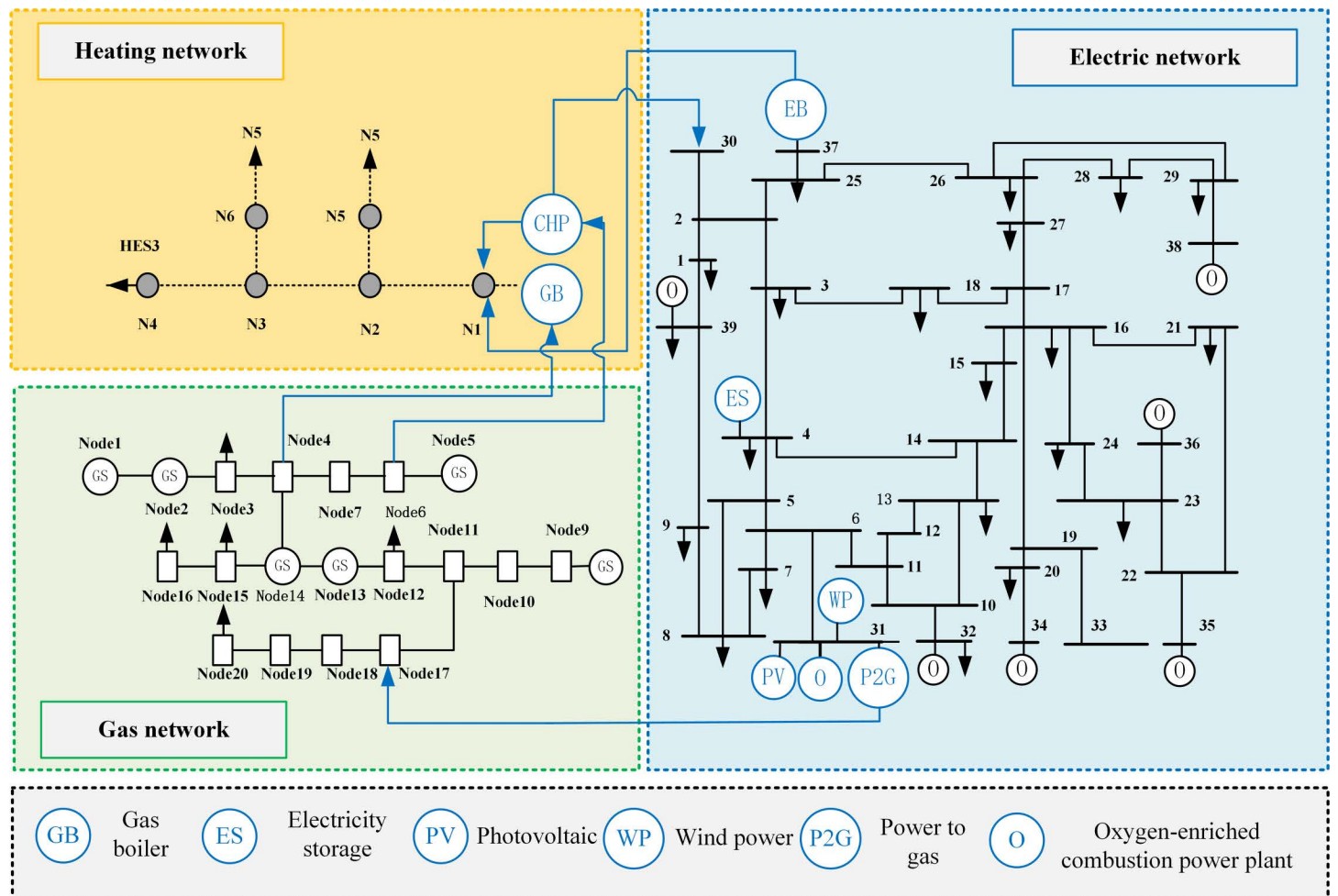

**Fig 6. 39-20-6 Integrated Energy System.**

In order to assess the validity and efficacy of the low-carbon economic dispatch strategy for the IES under the conditions of considering OCPP-P2G-CHP coupling proposed in this chapter, five operation scenarios are set up for low-carbon optimal dispatch, and the setup scenarios are as follows:

Scenario 1: The unit is a conventional thermal power unit, CC is not considered, and DR is not considered.

Scenario 2: The thermal power unit is reformed with oxygen combustion technology, the operation mode under the OCPP-P2G-CHP coupling condition is not considered, and DR is not considered.

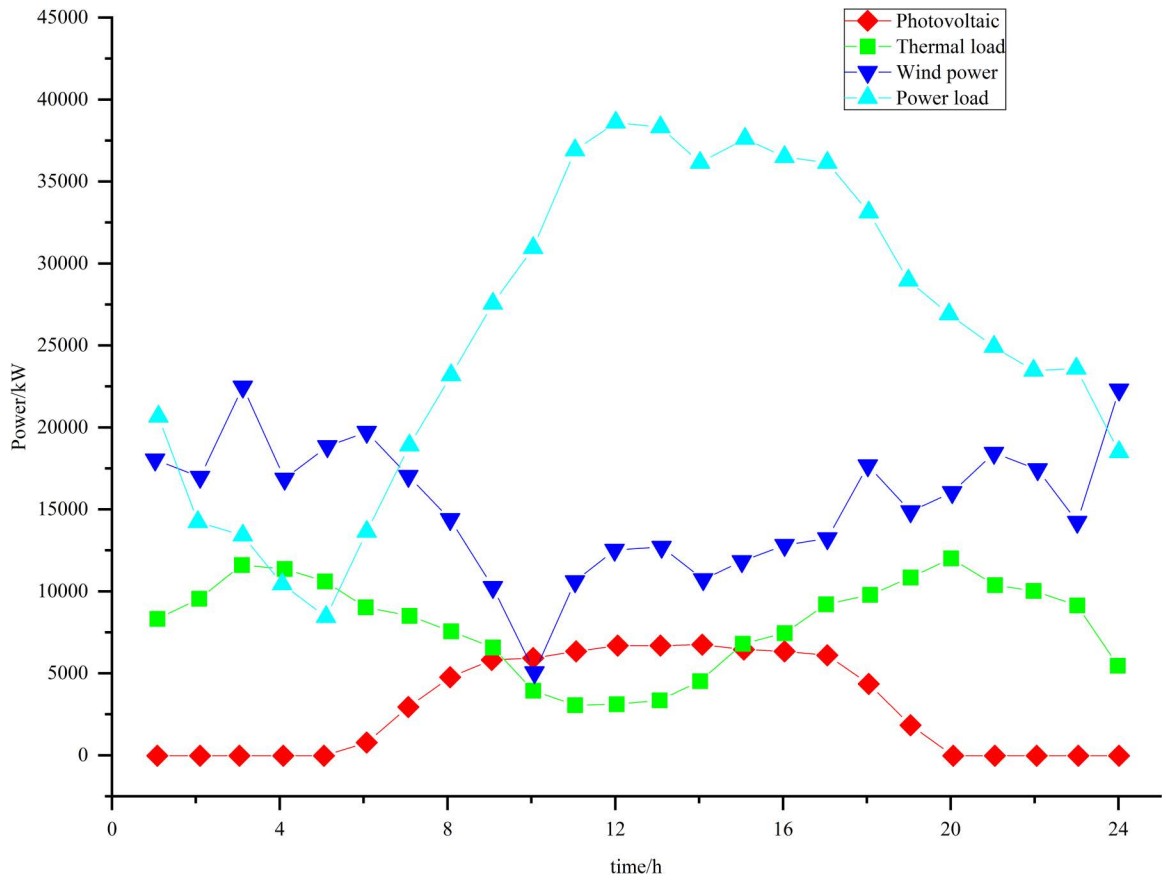

**Fig 7. Forecasts of wind, photovoltaic, electric and thermal load.**

Scenario 3: The thermal power unit is reformed with oxygen combustion technology, the operation mode under the OCPP-P2G-CHP coupling condition is not considered, and DR is considered.

Scenario 4: The thermal power unit undergoes oxygen combustion technology reform, the operation mode under the OCPP-P2G-CHP coupling condition is considered, and DR is not considered.

Scenario 5: The thermal power unit undergoes oxygen combustion technology reform, the operation mode under the OCPP-P2G-CHP coupling condition is considered, and DR is considered.

### 4.2. Comprehensive comparative analysis

Comprehensive comparative analysis shows (Table 4):

(1) Scenario 5 enhances the IES in comparison to Scenario 1, addressing both technical aspects (specifically through the implementation of oxygen-enriched combustion technology) and policy considerations (notably via a demand response mechanism). This scenario achieves a reduction in carbon emissions by 30.2% relative to Scenario 1. Additionally, a portion of the captured carbon dioxide is utilized as feedstock for P2G processes, resulting in a 26.1% decrease in carbon emission costs. Consequently, the overall system cost is reduced by 7.8% compared to Scenario 1. These findings indicate that Scenario 5 is more effective in simultaneously decreasing carbon emissions and lowering system costs.

 

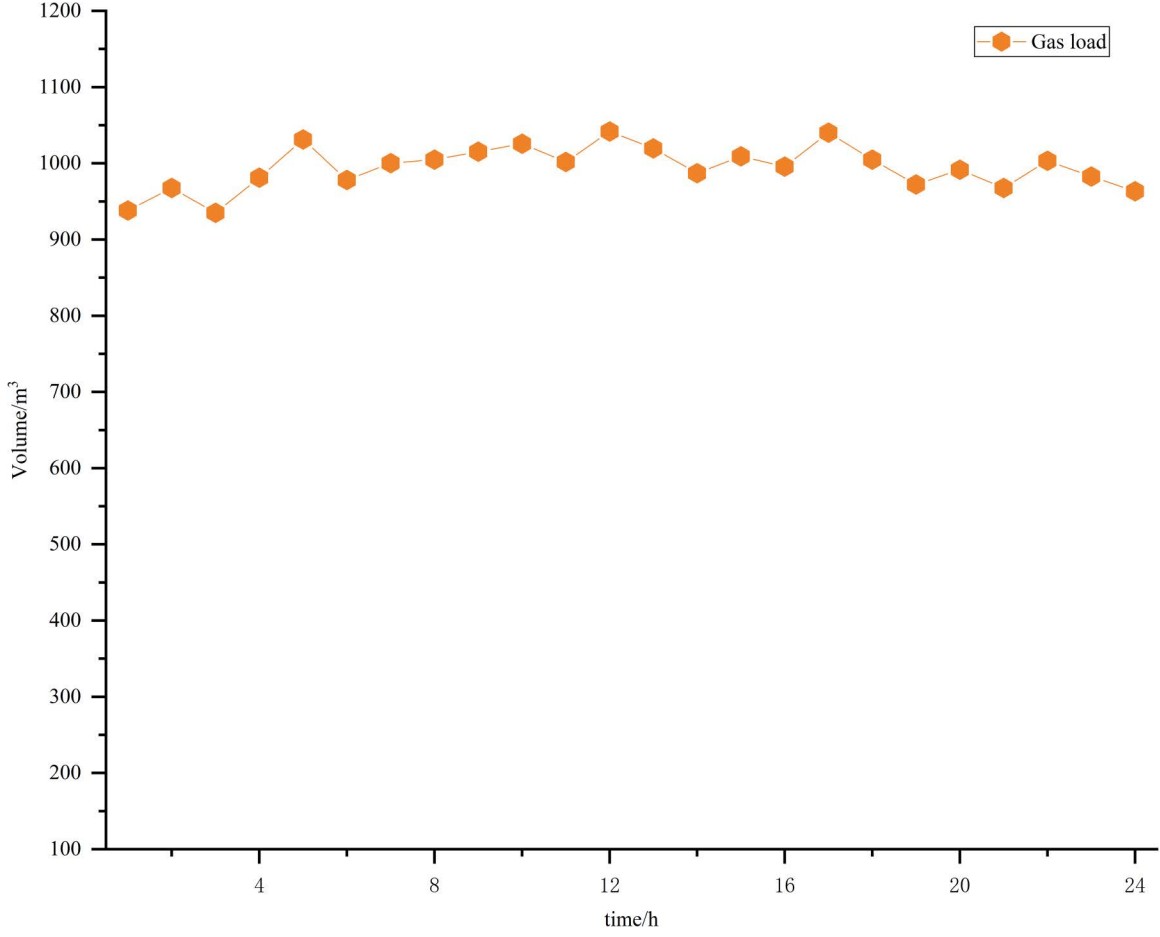

**Fig 8. Gas load forecast.**

**Table 4. Comparison of Scenarios.**

| Scenario | Total cost/¥ | Cost of purchasing energy/¥ | Coal consumption cost/¥ | Demand response Cost/¥ | Operating cost/¥ |
|---|---|---|---|---|---|
| 1 | 159282.45 | 55897.76 | 62453.22 | 0 | 10355.70 |
| 2 | 157741.31 | 45732.54 | 75398.76 | 0 | 11573.21 |
| 3 | 148353.50 | 33453.90 | 73378.23 | 8489.66 | 11531.16 |
| 4 | 155843.73 | 46630.55 | 74928.21 | 0 | 11248.76 |
| 5 | 146762.66 | 35350.17 | 71092.09 | 8276.40 | 11467.35 |
| Scenario | Wind and solar abandonment cost/¥ | Carbon emission cost/¥ | Carbon sequestration cost/¥ | Carbon emission/t | Carbon capture/t |
| 1 | 4593.12 | 25982.65 | 0 | 122.86 | 0 |
| 2 | 2047.87 | 21456.40 | 1532.53 | 113.87 | 40.56 |
| 3 | 0 | 20232.15 | 1268.40 | 99.76 | 38.43 |
| 4 | 1534.40 | 19745.65 | 1756.16 | 87.65 | 53.65 |
| 5 | 0 | 19211.50 | 1365.15 | 75.77 | 42.44 |

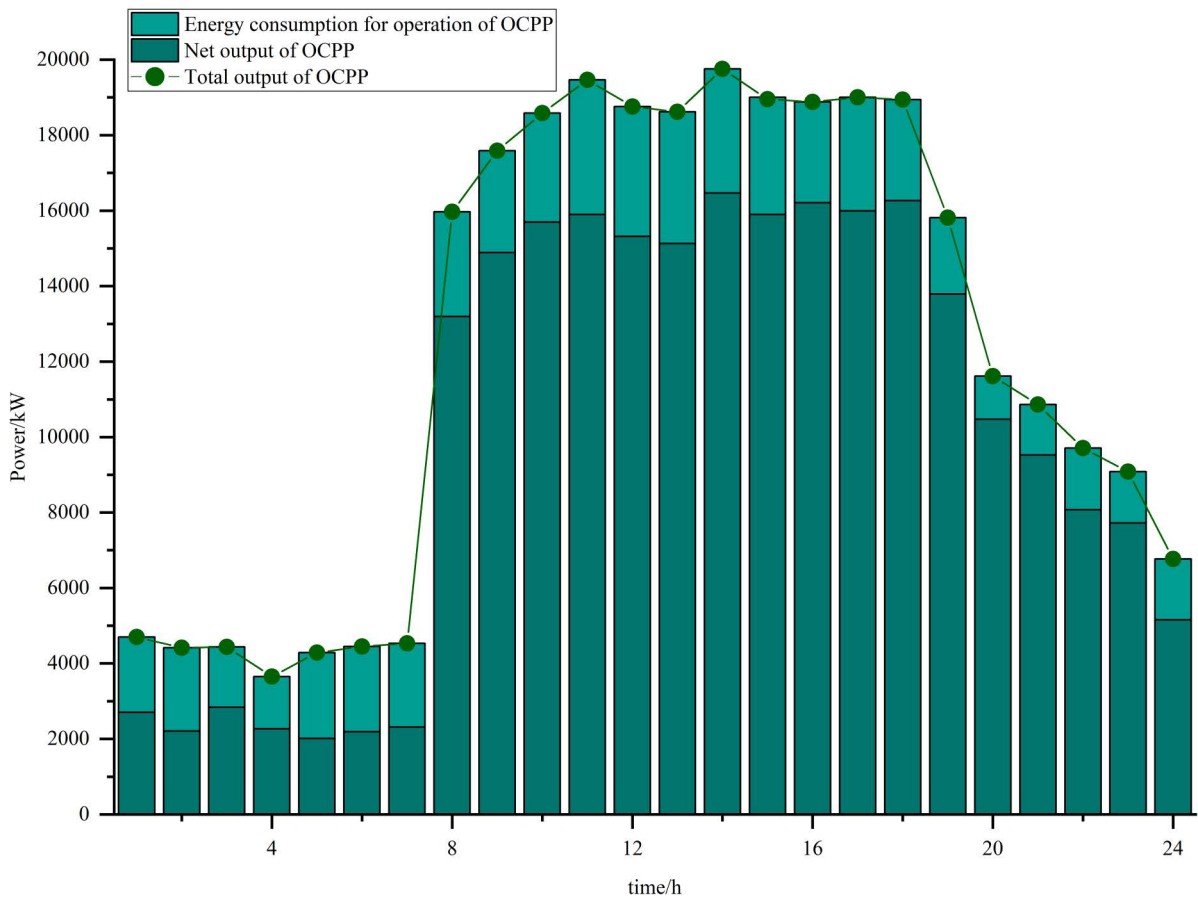

**Fig 9. The output of the OCPP.**

(2) Scenario 5 incorporates a demand response mechanism into Scenario 4. In comparison to Scenario 4, Scenario 5 demonstrates a reduction of 5.8% in total system costs and a 13.6% decrease in carbon emissions. Similarly, Scenario 3 introduces a demand response mechanism to Scenario 2, resulting in a 6.0% reduction in total system costs and a 12.4% decrease in carbon emissions relative to Scenario 2. Demand response refers to the practice whereby electricity consumers adjust their consumption patterns in response to price signals or incentives, thereby influencing total system costs and carbon emissions. This mechanism contributes positively to the stability, economic efficiency, and various other dimensions of the power system.

(3) In Scenario 5, the energy requirements for the carbon capture operation are distributed between the OCPP and the CHP system. Conversely, in Scenario 3, each power supply unit functions autonomously, with the energy necessary for the carbon capture operation being solely provided by the OCPP. Additionally, in Scenario 5, the $CO_2$ emissions from the cogeneration unit are captured by the carbon capture unit associated with the OCPP, which subsequently supplies the $CO_2$ required for P2G processes. As a result, when compared to Scenario 3, Scenario 5 demonstrates a reduction in total system costs by 1.1%, an increase in carbon capture by 4.01 tons, a decrease in demand response costs by 2.5%, and a reduction in carbon emissions by 24.0%. The CHP's provision of operational energy to both the OCPP and the carbon capture facility facilitates the P2G's supply of the oxygen needed by the OCPP, thereby

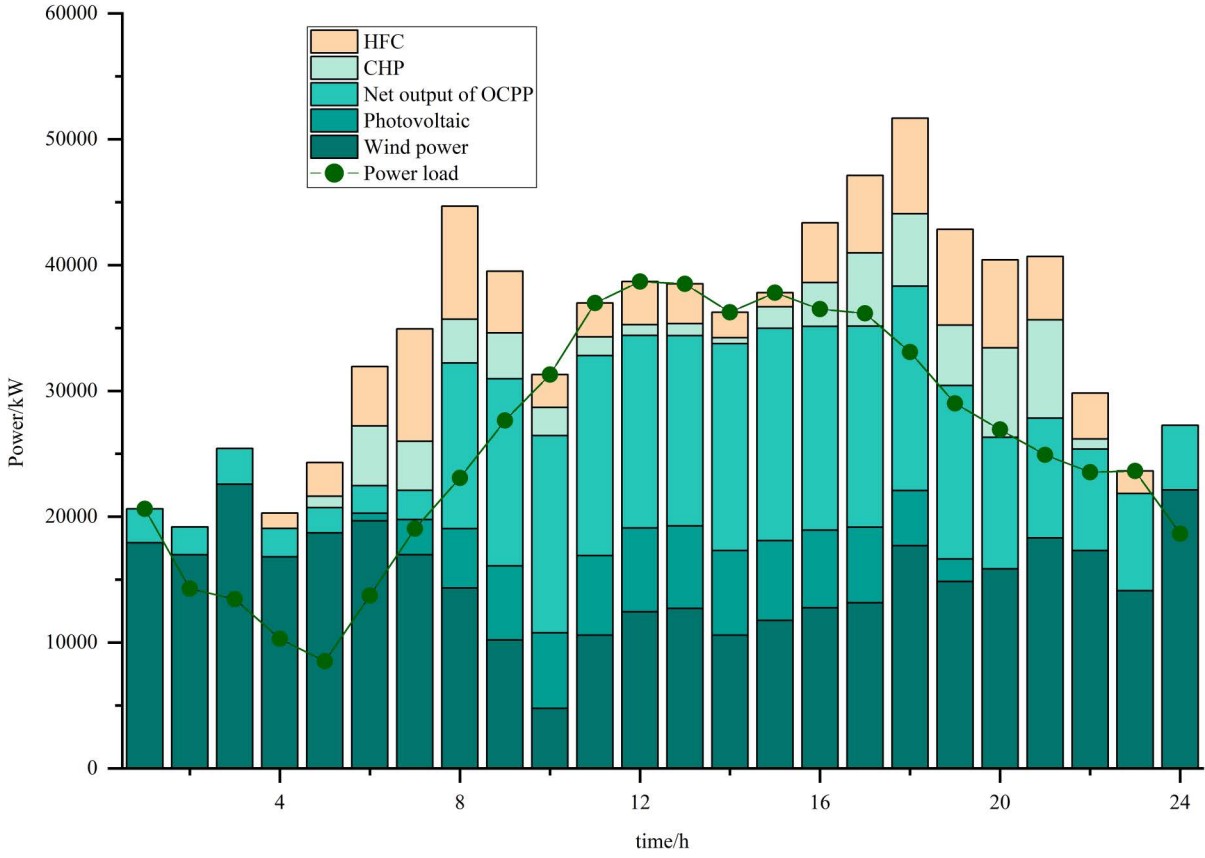

**Fig 10. Power supply for each power supply unit.**

conserving energy for the ASU and ultimately lowering the overall energy consumption of the OCPP, which in turn diminishes coal consumption costs. The "OCPP-P2G-CHP coupling" model presented in this study effectively synchronizes the output of each power generation unit, enhancing dispatch flexibility and demonstrating superior performance in terms of carbon reduction and economic efficiency compared to the independent operation model of each power supply unit.

In conclusion, this study examines the low-carbon economic dispatch model of IES under the coupling condition of OCPP-P2G-CHP. This model demonstrates significant benefits in terms of economic efficiency and reduced carbon emissions.

### 4.3. Analysis of optimized dispatch results

Simulations are carried out based on the IES low-carbon economic dispatch model considering the OCPP-P2G-CHP coupling condition proposed in this paper. In Scenario 5, the output of the OCPP, the power supply for each power supply unit, the thermal supply for each thermal supply unit, the natural gas power balance, the provision of energy for carbon capture, the carbon capture and emissions and the Hydrogen power balance, are illustrated in Figs 9–15.

As illustrated in Fig 9, there exists a significant disparity between the gross output power and net output power of the oxygen-enriched combustion power plant during peak power load hours, specifically from 10:00–18:00. This discrepancy

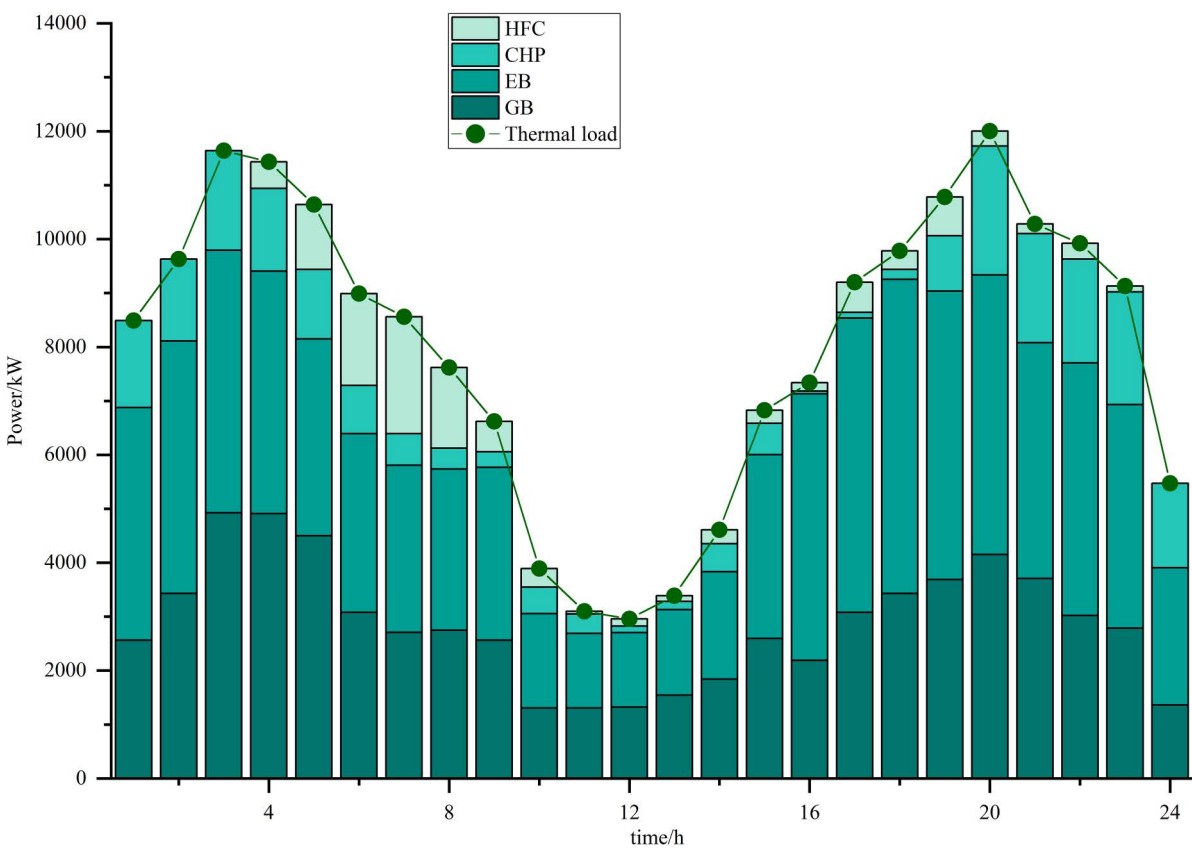

**Fig 11. Thermal supply for each thermal supply unit.**

is primarily attributed to the substantial energy consumption associated with oxygen preparation by the ASU and the carbon capture processes conducted by the CPU. The OCPP has the capability to leverage the oxygen provided by the OST to enhance the oxygen-enriched environment in instances where there is inadequate preparation of O2, thereby ensuring that the efficacy of carbon capture is maintained. The OCPP operates at a significantly elevated output during the hours of 08:00–17:00, a period characterized by diminished wind power generation and heightened energy demand. Conversely, during the hours of 20:00–24:00, when wind energy production is more abundant and energy demand is reduced, the OCPP is adjusted to function within a lower power range to optimize the utilization of available wind energy.

As illustrated in Fig 10, the periods from 00:00–05:00 and from 22:00–24:00 represent the peak times for renewable energy generation, with wind power contributing the majority of the electricity produced. The process of P2G facilitates the conversion of surplus electricity, thereby mitigating the electric-thermal coupling characteristics associated with CHP systems. Conversely, the period from 16:00–21:00 is characterized as a low-peak period for renewable energy, during which there is an increase in electricity demand. To maintain a stable power supply during this time, the output from both CHP and HFC is augmented.

As illustrated in Fig 11, the thermal power utilized to meet the thermal load is supplied by GB, EB, CHP, and HFC. During periods of elevated renewable energy utilization, the CHP systems generate heat energy. Conversely, in times of reduced renewable energy availability, when wind and photovoltaic sources struggle to meet the electric load demands, the CHP systems produce electrical power to maintain the reliability of the overall power supply. During these instances, the thermal power is predominantly supplied by GB and HFC systems.

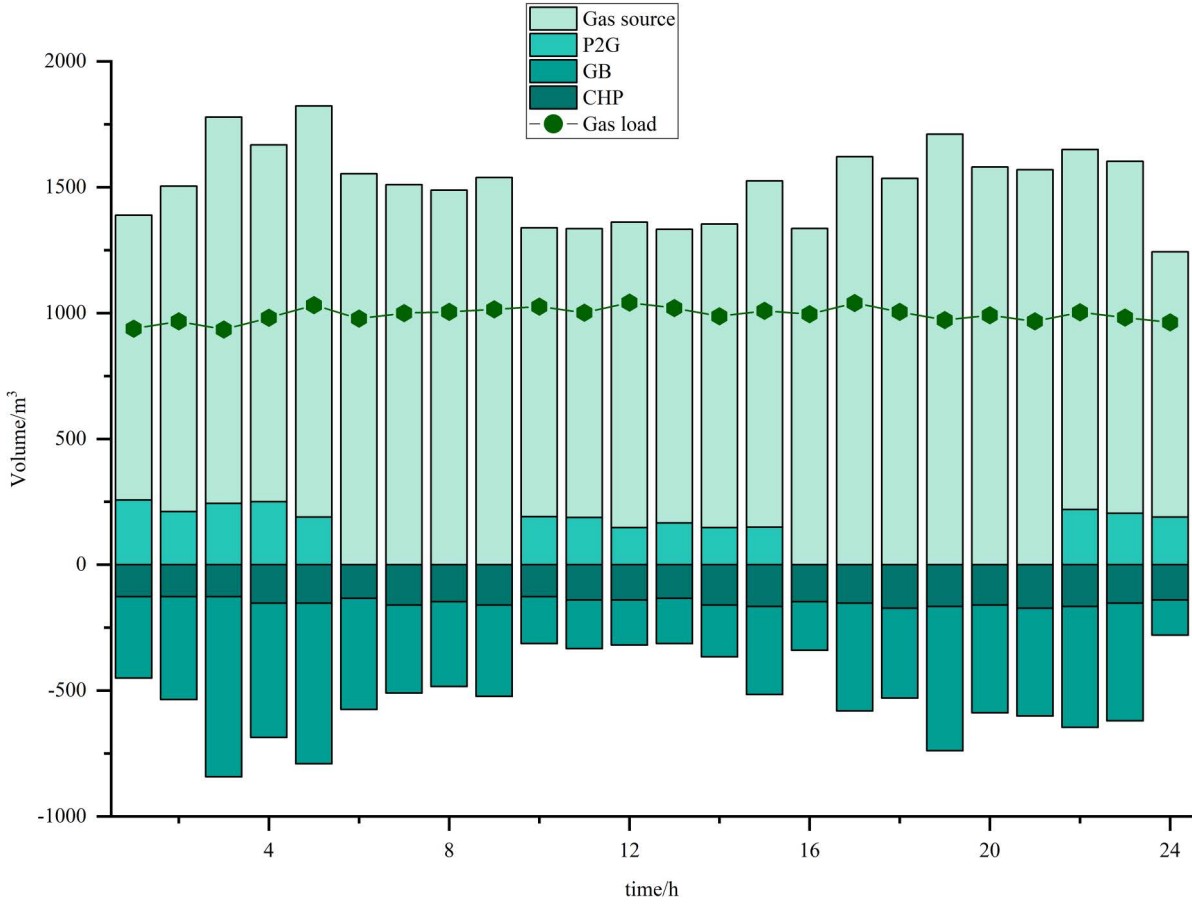

**Fig 12. Natural gas power balance.**

As illustrated in Fig 12, during the peak periods of renewable energy generation, P2G technology facilitates the conversion of electrical power into gas, thereby supplying the gas requirements for both the GB and the gas load. During this phase, the output power from the gas source is diminished, leading to a reduction in operational costs, a decrease in gas procurement expenses, and an enhancement in the overall economic efficiency of the system's operation. Notably, the GB exhibits increased gas consumption during the time intervals of 01:00–06:00 and 19:00–23:00. This phenomenon can be attributed to the peak thermal load occurring during these periods, coinciding with a lower availability of renewable energy sources, which necessitates reliance on the GB as the primary thermal supply.

Analysis of Figs 13 and 14 indicates that between the hours of 02:00 and 08:00, a period characterized by low power load, the energy consumption associated with carbon capture is minimal. This results in a correspondingly low volume of $CO_2$ captured. During this timeframe, the output power of the OCPP is also reduced, with the energy required for carbon capture primarily supplied by the CHP system. Conversely, from 15:00–20:00, when power load peaks, the output power of the OCPP increases while that of the CHP decreases. Consequently, the energy demands for carbon capture operations are predominantly met by the OCPP during this period. This shift leads to a substantial enhancement in carbon capture capacity, enabling the effective capture of the majority of $CO_2$ emissions produced, thereby contributing to a reduction in overall carbon emissions.

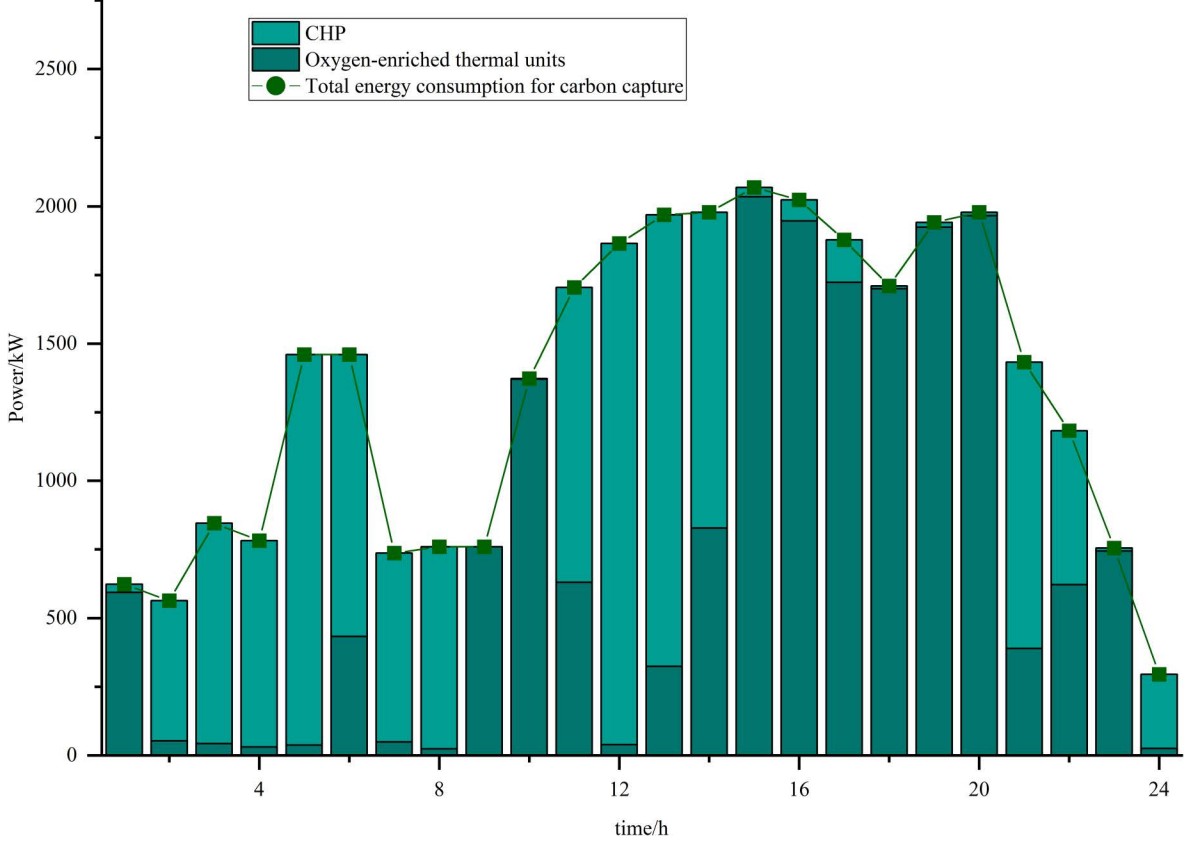

**Fig 13. The provision of energy for carbon capture.**

As illustrated in Fig 15, the hydrogen generated during the initial phase of the P2G electrolyzer between the time intervals of 1:00–2:00, 6:00–16:00, and 23:00–24:00 is entirely stored by the HFC and subsequently converted into electrical and thermal energy, without yielding any additional power output. Conversely, during the intervals of 3:00–5:00 and 18:00–22:00, a portion of the surplus hydrogen is produced to facilitate the P2G facility's conversion of hydrogen into natural gas. This phenomenon occurs due to elevated wind power generation coupled with reduced energy demand during these time frames, allowing the hydrogen produced by the electrolyzer to be utilized for natural gas production, thereby mitigating a portion of the system's energy procurement costs.

## 5. Conclusions

In this paper, the OCPP-P2G-CHP coupled IES low-carbon economic dispatch model is established on the basis of the IES of power-thermal-gas. The following conclusions are drawn from the simulation analysis of the example:

(1) The carbon capture technology utilizing oxygen-enriched combustion, as presented in this study, has the potential to substantially diminish carbon emissions within the system. Furthermore, the synergistic integration of this technology with P2G and CHP systems facilitates the recycling and reutilization of carbon resources, thereby enhancing the system's low-carbon operational efficiency. This evidence supports the assertion that oxygen-enriched combustion offers a beneficial approach to low-carbon economic dispatching.

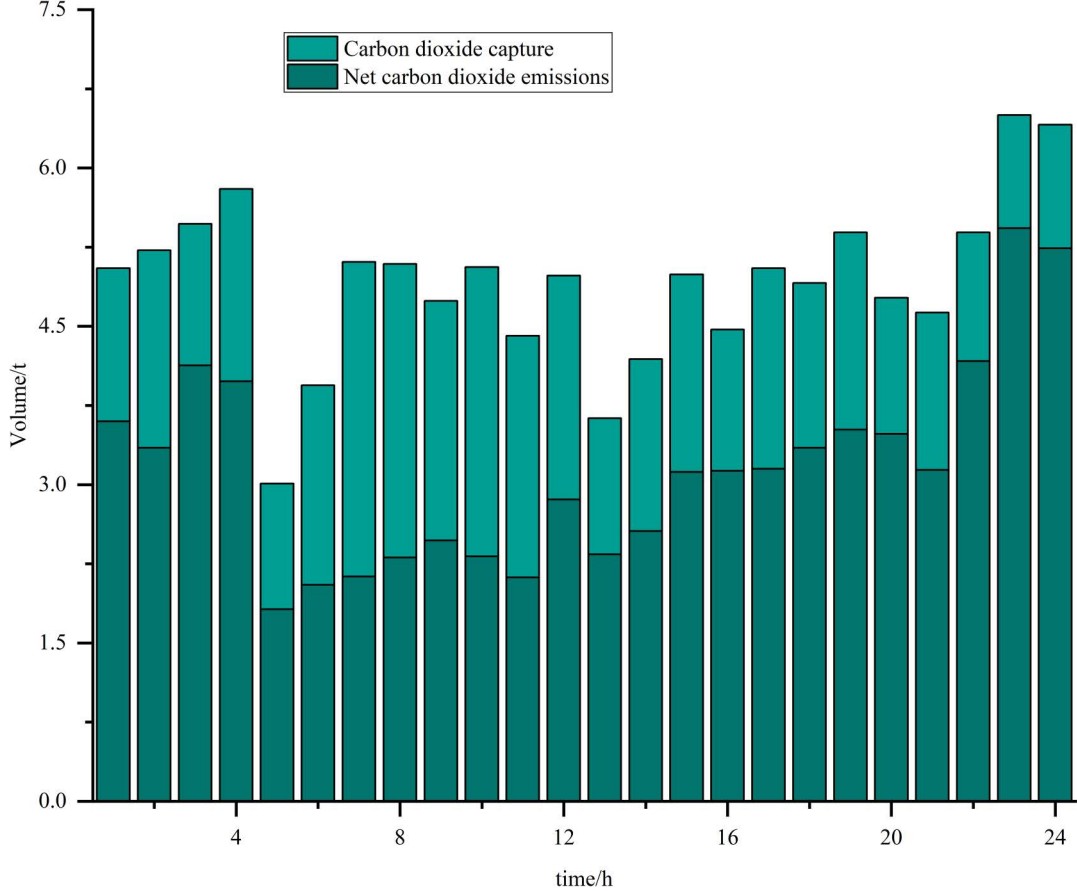

**Fig 14. The carbon capture and emissions.**

(2) The integrated operation of OCPP-P2G-CHP presented in this study demonstrates that the CHP system supplies surplus output to the OCPP, thereby enhancing the performance of the carbon capture unit. Furthermore, this integration alleviates the operational limitations of the CHP system and facilitates carbon recycling, leading to a decrease in coal expenses and the overall operational costs of the OCPP. Specifically, the total system cost is reduced by 1.1%, while coal costs are diminished by 3.12%.

(3) The coupled OCPP-P2G-CHP operation presented in this study demonstrates significant advantages over conventional scheduling methods. However, the analysis overlooks the consideration of the total life cycle cost, and certain aspects of the model fail to account for the effects of dynamic fluctuations on dispatching outcomes. Future investigations should prioritize the dynamic optimization of the model as well as the optimization of total life cycle costs.

## Supporting information

**S1 Fig. Wind forecast.**

(TIF)

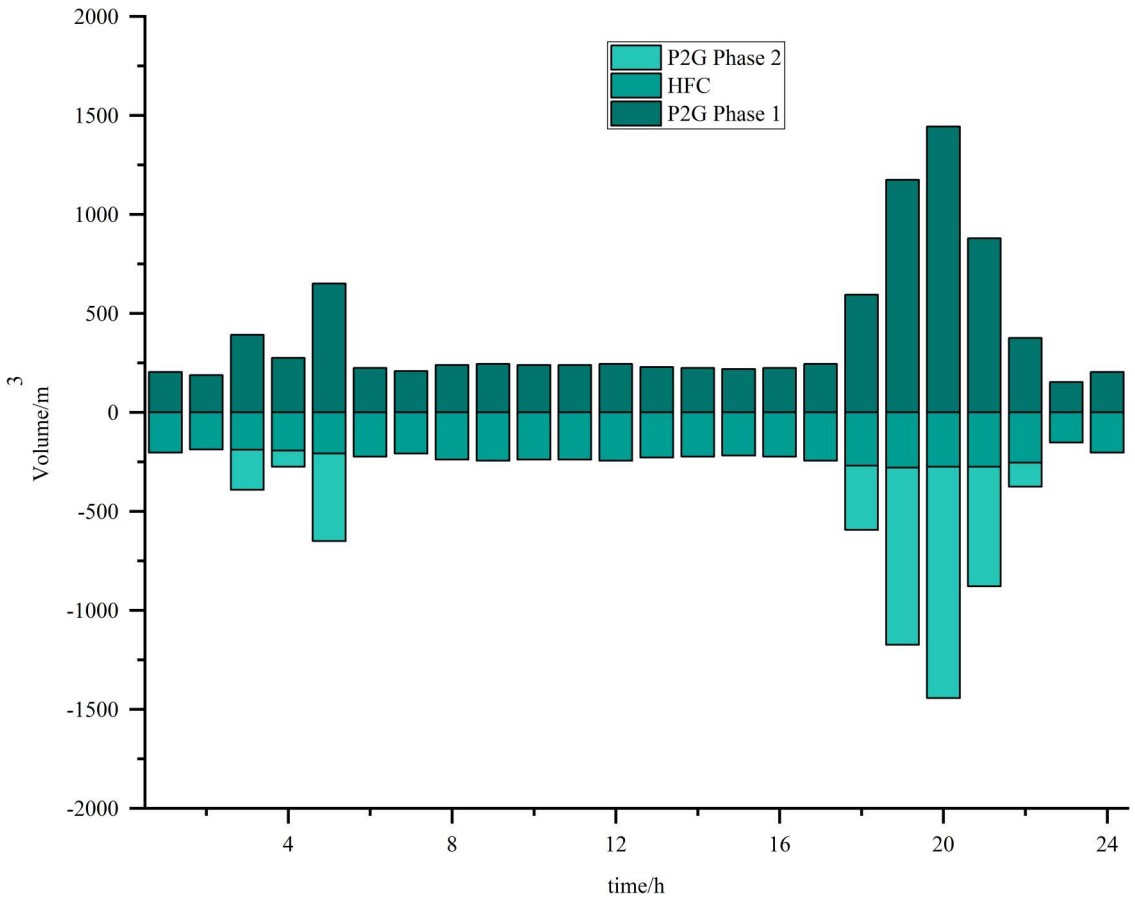

**Fig 15. Hydrogen power balance.**

**S2 Fig. Photovoltaic forecast.**
(TIF)

**S3 Fig. Electric load forecast.**
(TIF)

**S4 Fig. Heat load forecast.**
(TIF)

**S1 File. Forecast data.**
(XLSX)

## Author contributions

**Data curation:** Jingjing Ma.

**Formal analysis:** Jingjing Ma, Wentao Huang, Fan Liu.

**Investigation:** Jingjing Ma.

**Software:** Jingjing Ma, Can Lv, Fan Liu, Jun He.

**Writing – original draft:** Jingjing Ma, Can Lv, Jun He.

**Writing – review & editing:** Jingjing Ma, Wentao Huang, Yukun Yang.

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
