## [Decision Letter · Decision Letter 0]

18 Feb 2025

PONE-D-25-03463Low Carbon Economic Dispatch of Integrated Energy System Based on Coupled Operation of OCPP-P2G-CHPPLOS ONE

Dear Dr. Ma,

Thank you for submitting your manuscript to PLOS ONE. After careful consideration, we feel that it has merit but does not fully meet PLOS ONE’s publication criteria as it currently stands. Therefore, we invite you to submit a revised version of the manuscript that addresses the points raised during the review process.

We look forward to receiving your revised manuscript.

Kind regards,

Soheil Mohtaram

Academic Editor

PLOS ONE

Journal Requirements:

Reviewers' comments:

Reviewer's Responses to Questions

**Comments to the Author**

1. Is the manuscript technically sound, and do the data support the conclusions?

Reviewer #1: Yes

Reviewer #2: Yes

2. Has the statistical analysis been performed appropriately and rigorously? 

Reviewer #1: Yes

Reviewer #2: N/A

3. Have the authors made all data underlying the findings in their manuscript fully available?

Reviewer #1: Yes

Reviewer #2: Yes

4. Is the manuscript presented in an intelligible fashion and written in standard English?

Reviewer #1: Yes

Reviewer #2: Yes

5. Review Comments to the Author

Reviewer #1: This paper proposes an OCPP-P2G-CHP coupled operation mode, transforming the traditional thermal power unit into an oxygen-enriched combustion power plant (OCPP), and establishing the mathematical model of OCPP, two-stage P2G and CHP and their coupling mechanism to construct an integrated energy system (IES) model. Five scenarios are designed to analyze the output and carbon emissions of each subsystem, and the scheduling results of each scenario are compared to verify the feasibility and advantages of the proposed model.

1. The various parts of the model mentioned in the paper are closely related, but all of them are modeled using fixed-value linear models, without considering the dynamic characteristics of the model.

2. The following related research can be compared a) Two-Stage Coordinated Operation of A Green Multi-Energy Ship Microgrid With Underwater Radiated Noise by Distributed Stochastic Approach b) A multi-stage stochastic dispatching method for electricity-hydrogen integrated energy systems driven by model and data

3. The innovation of the article is not strong enough. The oxygen-enriched combustion power plant (OCPP) is the core of the innovation of this paper, but the modeling is too rough. Please enrich the modeling of this part.

4. The simulation part of this paper lacks the power system topology diagram and the photovoltaic and wind power data used lacks source citations.

5. Scheduling is the title of this paper but the simulation part is too rough. One hour as the scheduling step is a bit too long for this model. Please reconsider the scheduling step.

6. Figures 2, 3, 5 and 6 in this paper are too rough. Please redraw them.

Reviewer #2: The paper deals with the low-carbon economic dispatch topic by proposing and investigating the performances of an optimization algorithms for an integrated power and thermal energy system including OCPP, P2G, and CHP facilities. Noteworthy the very deeply demonstration of the mathematical approach and the excellent English style. The simulation results of the proposed approach show better performances against other energy system models. The paper presents a significant contribution on the topic offering a new perspective on the addressed problem, the obtained results are interesting and reliable. Other strengths of the paper refer to the logical structure of ideas, clear presentation of the research purpose and objectives, and well organization oh the paper content. Moreover, the methodology is described in detail, allowing other researchers to reproduce the experiment; the results are presented clearly and concisely, by using relevant graphs and tables, and the analysis of the results is done in-depth.

However, the manuscript should be revised as there are several issues to be addressed for the sake of clarity and simplicity.

Minor issues

1. Abstract: what is the novelty addressed by your research? Please state explicitly the novelty involved in your approach. Quantitative data/conclusions are also welcome.

2. Avoid repeating frequently an acronym and its definition. E.g., Integrated Energy Systems (IES).

3. Recommendation to include in the paper a nomenclature of the used symbols & acronyms

4. Proposal to consider the much significant syntagma “Author et al. [x]...” instead of anonymous “In [x]…”, “Ref [x] presents”, etc.

5. Solve several typing mistakes, e.g. “processes. [26].”, “is the Net”, “boilers(GB)modeling”, “boilers(EB)”, etc. Check carefully the entire manuscript for other similar mistakes.

6. Typically, the paper structure (its sections) is briefly presented at the end of Introduction.

7. Define all symbols used in equations and/or include their definition / significance in Nomenclature.

8. Increase the quality (resolution) of all figures, they are in current state at the legibility limit.

9. Fig. 2: the role of the CC unit in the OCPP system in not clearly explain (it is isolated form the rest of the components, there is only a power input, without a specific output). Similarly for other figures, where the local outputs should be also indicated.

10. Justify the selection of curves in Fig. 5 and Fig. 6: what they represent in this specific case study? Are they coming from measurements? Please clarify it!

Major issues

11. The literature review: an appropriate systematization of the identified solutions and challenges in the addressed topic is more valuable instead of presenting one by one the paper results.

12. A flowchart of the proposed approach is welcomed before introducing all mathematical issues.

13. Use appropriate reference(s) whenever unproven equations are introduced (i.e., the large majority of the equations). Make a clear difference between existing and proposed equations.

14. Objective function: some clarification are needed related to the cost optimization concept by considering the whole life cycle cost of the system, not only the operating cost. Or, at least state the limits of the proposed approach.

15. 6. Conclusion: proposal to rename it “6. Conclusions” as several conclusions are drawn. The limits of the proposed system and approach, as well as future works, should be highlighted here.

6. PLOS authors have the option to publish the peer review history of their article (what does this mean? ). If published, this will include your full peer review and any attached files.

**Do you want your identity to be public for this peer review?** For information about this choice, including consent withdrawal, please see our Privacy Policy .

Reviewer #1: No

Reviewer #2: No

---

## [Author Response · Author response to Decision Letter 1]

26 Feb 2025

Responds to the reviewer’s comments:

Reviewer #1:

1. The various parts of the model mentioned in the paper are closely related, but all of them are modeled using fixed-value linear models, without considering the dynamic characteristics of the model.

The author's answer: Thank you very much for this suggestion.We are very sorry for our negligence of the dynamic characteristics of the model. We have made corrections based on your comments. We modified the model for OCPP, CHP, and P2G, and added climbing constraints on the basis of the original model in order to enhance the model's portrayal of the dynamic process and reflect the dynamic regulation capability of the core technology of this paper more realistically.

2. The following related research can be compared a) Two-Stage Coordinated Operation of A Green Multi-Energy Ship Microgrid With Underwater Radiated Noise by Distributed Stochastic Approach b) A multi-stage stochastic dispatching method for electricity-hydrogen integrated energy systems driven by model and data.

The author's answer: Considering your suggestion, we have compared these studies and placed them in the literature review of this paper. We refer to these papers as references 18 and 19.

3. The innovation of the article is not strong enough. The oxygen-enriched combustion power plant (OCPP) is the core of the innovation of this paper, but the modeling is too rough. Please enrich the modeling of this part.

The author's answer: We deeply apologize for the shortcomings in the modeling of OCPP. Following your suggestions, we have supplemented and enriched the original model and added some textual narratives, which we hope will meet your requirements.

4. The simulation part of this paper lacks the power system topology diagram and the photovoltaic and wind power data used lacks source citations.

The author's answer: We apologize for the lack of power system topology diagrams. Therefore, we have added the power system topology diagram in the simulation section, labeled as Fig. 6, and the energy flow corresponds to Fig. 2. Forecasts for wind, photovoltaic, and loads are from Ref. 40, with cited sources noted in the article.

5. Scheduling is the title of this paper but the simulation part is too rough. One hour as the scheduling step is a bit too long for this model. Please reconsider the scheduling step.

The author's answer: We are equally in strong agreement about the simulation portion of the problem, and we are acutely aware of the issues regarding the steps of the program. However, due to the limitations of the data and literature on wind, photovoltaic and load forecasting, we found it difficult to undertake this work because it was beyond the scope of our simulations. To address the roughness of the simulation part, we added the hydrogen power balance, labeled Figure 15, to the analysis of the scheduling results and analyzed it. The above deficiencies are also the direction of our future thesis research. We hope to have your understanding.

6. Figures 2, 3, 5 and 6 in this paper are too rough. Please redraw them.

The author's answer: Taking into account the reviewers' suggestions for the figures, we have re-processed the figures for clarity. We apologize to the reviewers for any reading difficulties.

Reviewer #2:

1. Abstract: what is the novelty addressed by your research? Please state explicitly the novelty involved in your approach. Quantitative data/conclusions are also welcome.

The author's answer: Thank you for this suggestion.Based on your suggestions for the abstract, we have revised it to better convey the innovative contribution and practical significance of the article.

2. Avoid repeating frequently an acronym and its definition. E.g., Integrated Energy Systems (IES).

The author's answer: We sincerely apologize for our oversight. We appreciate your reminder, which prompted us to review the entire text and implement the necessary corrections.

3. Recommendation to include in the paper a nomenclature of the used symbols & acronyms.

The author's answer: Your suggestion is very rigorous and reasonable.The manuscript has been supplemented with an acronym nomenclature. However, due to the extensive number of specialized symbols utilized within the article, providing a comprehensive list would be excessively space-consuming. Consequently, we have opted not to include a detailed explanation of each symbol; instead, we have clarified the meaning of the symbols upon their initial occurrence in the text. We regret this decision and kindly request the reviewers' understanding.

4. Proposal to consider the much significant syntagma “Author et al. [x]...” instead of anonymous “In [x]…”, “Ref [x] presents”, etc.

The author's answer: We express our sincere appreciation to the reviewers for their valuable suggestion. Consequently, we have implemented grammatical revisions in the literature review section of the manuscript.

5. Solve several typing mistakes, e.g. “processes. [26].”, “is the Net”, “boilers(GB)modeling”, “boilers(EB)”, etc. Check carefully the entire manuscript for other similar mistakes.

The author's answer: Thanks for your careful checks.We feel sorry for our carelessness. We have corrected it and we also feel great thanks for your point out. We apologize for any reading difficulties you may have had!

6. Typically, the paper structure (its sections) is briefly presented at the end of Introduction.

The author's answer: Thank you for this suggestion, for which we give a brief description of the structure of the paper at the end of the introduction.

7. Define all symbols used in equations and/or include their definition / significance in Nomenclature.

The author's answer: Thank you for this suggestion, and again, for the sake of having too many symbols, we've only described them below the first appearance of the symbol. We hope you understand, and we apologize for this.

8. Increase the quality (resolution) of all figures, they are in current state at the legibility limit.

The author's answer: We apologize for the image clarity issue. For this reason, we have improved the clarity of all the images in the article to ensure the smoothness of your subsequent reading. We hope this error has not caused you any reading displeasure.

9. Fig. 2: the role of the CC unit in the OCPP system in not clearly explain (it is isolated form the rest of the components, there is only a power input, without a specific output). Similarly for other figures, where the local outputs should be also indicated.

The author's answer: Thank you very much for this suggestion for our paper. To this end we have revised Figure 2 to clearly illustrate the role of the CC unit in the OCPP system, including its inputs and outputs. For clarity, a corresponding explanation has been added to the text. Similar changes have also been made to the others.

10. Justify the selection of curves in Fig. 5 and Fig. 6: what they represent in this specific case study? Are they coming from measurements? Please clarify it!

The author's answer: The photovoltaic power curve in Figure 5 shows the variation of solar power generation over time, usually peaking during the day and dropping to zero at night. The wind power curve shows the variation of wind power generation over time, which is usually affected by changes in wind speed and is highly volatile. The thermal load curve shows the variation of the system's demand for thermal energy, related to industrial processes, heating or cooling needs. The electric load curve shows the variation in the system's demand for electricity, usually associated with industrial, commercial, and residential use. The natural gas load curve in Figure 6 shows the variation in system demand for natural gas, which may be associated with power generation, heating, or industrial processes. The photovoltaic, wind, and electric load data in Figure 5 can be used to optimize the synergistic operation of renewable energy sources with fossil energy sources. The natural gas load data in Figure 6 can be combined with the heat load data in Figure 5 to optimize the operation of combined heat and power (CHP) systems. Forecasts for wind, photovoltaic, and loads are from Ref. 40, with cited sources noted in the article. The data is from a comprehensive park and is indicative and representative.

11. The literature review: an appropriate systematization of the identified solutions and challenges in the addressed topic is more valuable instead of presenting one by one the paper results.

The author's answer: In response to this suggestion of yours, we have revised the literature review to specify the resolution and contribution of the references.

12. A flowchart of the proposed approach is welcomed before introducing all mathematical issues.

The author's answer: Thank you very much for this suggestion, so we have added a flowchart of the proposed approach in the introduction section.

13. Use appropriate reference(s) whenever unproven equations are introduced (i.e., the large majority of the equations). Make a clear difference between existing and proposed equations.

The author's answer: We couldn't agree more with you on this suggestion. Therefore we have added references [32-38] near some of the equations to distinguish the existing equations from the proposed ones.

14. Objective function: some clarification are needed related to the cost optimization concept by considering the whole life cycle cost of the system, not only the operating cost. Or, at least state the limits of the proposed approach.

The author's answer: Thank you very much for your professional comments on our article. As you are concerned, this issue needs to be addressed. However, due to the limitations of the available information, we are not able to optimize the whole life cycle cost of the system. Therefore, we have explained this shortcoming of the article in the conclusion section. Thank you again for this very valuable suggestion, which provides us with a direction for future research.

15. 6. Conclusion: proposal to rename it “6. Conclusions” as several conclusions are drawn. The limits of the proposed system and approach, as well as future works, should be highlighted here.

The author's answer: We were really sorry for our careless mistakes. Thank you for your reminder. We have added at the conclusion the shortcomings of the research methodology and the directions for future research needed.

---

## [Decision Letter · Decision Letter 1]

2 Apr 2025

Low Carbon Economic Dispatch of Integrated Energy System Based on Coupled Operation of OCPP-P2G-CHP

PONE-D-25-03463R1

Dear Dr. Jingjing Ma,

We’re pleased to inform you that your manuscript has been judged scientifically suitable for publication and will be formally accepted for publication once it meets all outstanding technical requirements.

Kind regards,

Alessio Faccia

Academic Editor

PLOS ONE

---

## [Editor Report · Acceptance letter]

PONE-D-25-03463R1

PLOS ONE

Dear Dr. Ma,

I'm pleased to inform you that your manuscript has been deemed suitable for publication in PLOS ONE. Congratulations! Your manuscript is now being handed over to our production team.

Kind regards,

on behalf of

Dr. PLOS Manuscript Reassignment

Staff Editor

PLOS ONE